# AVSET-10M: An Open Large-Scale Audio-Visual Dataset with High Correspondence

**Xize Cheng**[1*]   **Ziang Zhang**[1*]   **Zehan Wang**[1*]   **Minghui Fang**[1]   **Rongjie Huang**[1]
**Siqi Zheng**[2]   **Ruofan Hu**[1]   **Jionghao Bai**[1]   **Tao Jin**[1]   **Zhou Zhao**[1] [†]
[1]Zhejiang University   [2]Alibaba Group
{chengxize,zhaozhou}@zju.edu.cn

## Abstract

Groundbreaking research from initiatives such as ChatGPT and Sora underscores the crucial role of large-scale data in advancing generative and comprehension tasks. However, the scarcity of comprehensive and large-scale audio-visual correspondence datasets poses a significant challenge to research in the audio-visual fields. To address this gap, we introduce **AVSET-10M**, a audio-visual high-corresponding dataset comprising 10 million samples, featuring the following key attributes: (1) **High Audio-Visual Correspondence**: Through meticulous sample filtering, we ensure robust correspondence between the audio and visual components of each entry. (2) **Comprehensive Categories**: Encompassing 527 unique audio categories, AVSET-10M offers the most extensive range of audio categories available. (3) **Large Scale**: With 10 million samples, AVSET-10M is the largest publicly available audio-visual corresponding dataset. We have benchmarked two critical tasks on AVSET-10M: audio-video retrieval and vision-queried sound separation. These tasks highlight the essential role of precise audio-visual correspondence in advancing audio-visual research. For more information, please visit https://avset-10m.github.io/.

## 1   Introduction

Scaling up significantly enhances performance in understanding [37, 4, 26] and generation [20, 19, 42] tasks across visual and language modalities. Inspired by the success of ImageNet [9] in visual research, some introduce the pioneering large-scale audio dataset, AudioSet [12], which comprises 2.1 million audio samples each manually annotated with fine-grained audio categories to advance automatic audio understanding. However, the annotation process in AudioSet primarily focuses on only audio labels, neglecting the audio-visual correspondence. To address the need for exploring temporal consistency between audio and video, researchers develop the VGGSound [6], which includes 200,000 samples with audio-visual correspondence. Leveraging this dataset, significant breakthroughs have been achieved in the audio-visual domain, including vision-queried sound separation [10] and vision-based audio synthesis [14, 43].

Meanwhile, the scale of vision-language datasets [35, 29, 44, 32, 40] has expanded dramatically, encompassing up to 100 million or even 1 billion samples. This expansion has facilitated a qualitative leap in understanding [37, 26] and generation [20] tasks within the vision and language fields,

---

*Equal contribution.
†Corresponding author.

Submitted to the 38th Conference on Neural Information Processing Systems (NeurIPS 2024) Track on Datasets and Benchmarks. Do not distribute.

Table 1: Comparison of different audio-video datasets. **AV-C** denotes the audio-visual correspondence. **# Class**: Number of audio categories. ACAV-100M$^{\dagger}$ does not filter out the voiceover.

| Datasets | Video | AV-C | #Class | #Clips | #Dur.(hrs) | #Avg Dur.(s) |
|---|---|---|---|---|---|---|
| DCASE2017 [28] | ✗ | ✗ | 17 | 57K | 89 | 3.9 |
| FSD [11] | ✗ | ✗ | 398 | 24K | 119 | 17.4 |
| AudioSet [12] | ✔ | ✗ | 527 | 2.1M | 5.8K | 10 |
| AudioScope-V2 [39] | ✔ | ✗ | - | 4.9M | 1.6K | 5 |
| ACAV100M[22]$^{\dagger}$ | ✔ | ✗ | - | 100M | 277.7K | 10 |
| HD-VILA-100M [44] | ✔ | ✗ | - | 103M | 371.5K | 13.4 |
| Panda-70M [8] | ✔ | ✗ | - | 70.8M | 166.8K | 8.5 |
| AVE [36] | ✔ | ✔ | 28 | 4K | 11 | 10 |
| VGGSound [6] | ✔ | ✔ | 309 | 200K | 550 | 10 |
| AVSET-700K (ours) | ✔ | ✔ | 527 | 728K | 2.0K | 10 |
| AVSET-10M (ours) | ✔ | ✔ | 527 | 10.9M | 30.4K | 10.3 |

enabling the development of intelligent large language models [37] and video generation technologies [5] that simulate real-world scenarios. In contrast, the scale of datasets that ensure audio-visual correspondence remains markedly limited, posing a constraint on advancements in audio-visual field.

To further expand the audio-visual corresponding dataset and promote research on audio-visual temporal consistency, we propose AVSET-10M, the first 10 million scale audio-visual corresponding dataset, along with AVSET-700K, a subset containing fine-grained audio annotations. In Table 1, we present a comparison among various existing audio and audio-visual datasets. Our dataset construction process includes four stages: (1) Data collection, (2) Audio-visual correspondence filtering, (3) Voice-over filtering, and (4) Sample recycling with sound separation. We select AudioSet [12], known for its fine-grained manual labeling of audio categories, as our initial data source and develop AVSET-700K with accurate audio labels. To increase the number of samples per audio category, we choose Panda-70M [8] as an additional data source, expanding AVSET-700K to 10 million audio-visual corresponding samples. Panda-70M processes long videos into multiple semantically coherent sub-segments, effectively preventing the mixing of sounds from different events. Previous filtering method [6] using visual classification models struggles to distinguish abstract sounds without fixed visual content, such as silence, thereby limiting the diversity of audio categories. To address this issue, we introduce a new filtering method based on audio-visual similarity [13], which significantly broadens the diversity of audio types. We employ an audio classification model [21] to filter out samples containing narration or background music that does not align with the visual content. As speech is commonly found in wild video data, which often results in the inadvertent filtering out of a substantial amount of audio samples containing voice-overs. This leads to the loss of many potentially useful and valuable samples across various audio categories. Thus, we further attempt to employ sound separation models [33] to recycle as many of these wasted samples as possible. From the initial 41 million samples, we filter 10 million audio-visual samples with high correspondence. Verification experiments demonstrate that our AVSET-700K provides more robust audio-visual correspondence than the previously used audio-visual corresponding dataset (VGGSound). Additionally, benchmarks of audio-video retrieval and vision-queried sound separation on AVSET-10M demonstrate it offers more research opportunities in the field of audiovisual studies.

## 2 Related Works

### 2.1 Audio-Visual Models

As multi-modal research progresses, the investigation [24, 31, 17] into the correlations between audio and visual modalities has advanced. Initially, researchers employ both audio and video data to provide semantically richer information, thereby improving video understanding and significantly enhancing performance in various video understanding tasks such as video question answering (VQA) [24, 2],

video captioning [31, 15, 16, 25], and video retrieval [23, 17, 3]. Following these developments, ImageBind [13] emerges as a pioneering project that successfully aligns audio and visual content, marking a significant step in exploring semantic alignment between these modalities. Building on this foundation, subsequent research has delved into more intricate interactions between audio and video, achieving milestones in vision-queried sound separation [10] and video dubbing [14]. However, while these methods have managed to align audio and visual content semantically, they often falter in maintaining temporal consistency. Recent innovations [27] have introduced audio-visual temporal consistency supervision loss to enhance the temporal alignment in video dubbing.

Despite these advancements, the limited availability of training data continues to pose a significant challenge, keeping the development of audio-visual temporal consistency at a rudimentary level. As a result, the understanding of visual content remains largely confined to the semantic level, which hampers the ability of models to accurately capture the audio-visual temporal consistency.

## 2.2 Audio-Video Dataset

Inspired by ImageNet [9], researchers [12] annotate a substantial audio dataset, consisting of 2.1 million audio samples, aimed at enhancing automatic audio comprehension. Although annotators are encouraged to consult video content to refine the accuracy of audio annotations, the dataset primarily focuses on precise audio annotations without additional measures to filter out audio-visual non-corresponding samples. This limits the exploration of audio-video consistency.

To investigate the audio-visual consistency, researchers [6] employ a visual model [30] to identify sound-producing objects in videos, resulting in the creation of VGGSound, a dataset comprising 200,000 audio-visual corresponding samples. However, this visual model proves effective only in scenes characterized by definite actions or visible objects. It struggles to handle abstract audio scenes such as silence and urban outdoors, even though there is indeed a correlation between these abstract audio and the visual content. This constraint limits the diversity of audio categories represented in VGGSound. To further scale up audio-visual datasets, ACAV100M [22] employs a clustering-based approach to filter data. However, it does not filter out voice-overs, resulting in the audio-visual correspondence of the final dataset being even worse than that of AudioSet. AudioScope V1/2 [38, 39] uses an unsupervised audio-video consistency prediction model to evaluate the audio-video matching score and screens 2,500 hours of video samples from YFCC100M [35]. Nevertheless, due to the limitations in prediction accuracy, the consistency between audio and video cannot be guaranteed, and there is still a significant amount of inconsistent audio-visual content in the dataset.

Although subsequent research introduces larger video datasets [44, 40, 7, 8], the primary focus remains on exploring the relationship between video and text, overlooking the audio-visual correspondence. To the best of our knowledge, our AVSET-10M represents the largest open audio-visual high-correspondence dataset currently available, containing 10 million data samples across 527 different audio categories. This dataset opens up more opportunities for research in the audio-video field.

## 3 AVSET-10M

### 3.1 Dataset Construction Pipeline

**Stage 1: Data Collection.** We select two different open-source datasets, AudioSet [12] and Panda-70M [8], as data sources. All videos are sourced from open-domain YouTube content. Since these datasets do not focus on audio-visual correspondence, they contain a substantial number of mismatched audio-visual samples. We utilize a sophisticated filtering process to select samples with high audio-visual correspondence, thereby constructing AVSET-10M.

AudioSet [12] is a pioneering large-scale audio dataset where all audio category labels are carefully annotated by human annotators. During the annotation process, annotators are allowed to view the accompanying videos, which aids in accurate audio category identification. This dataset includes 2.1 million audio samples across 527 unique audio categories. From AudioSet, we select 727,530

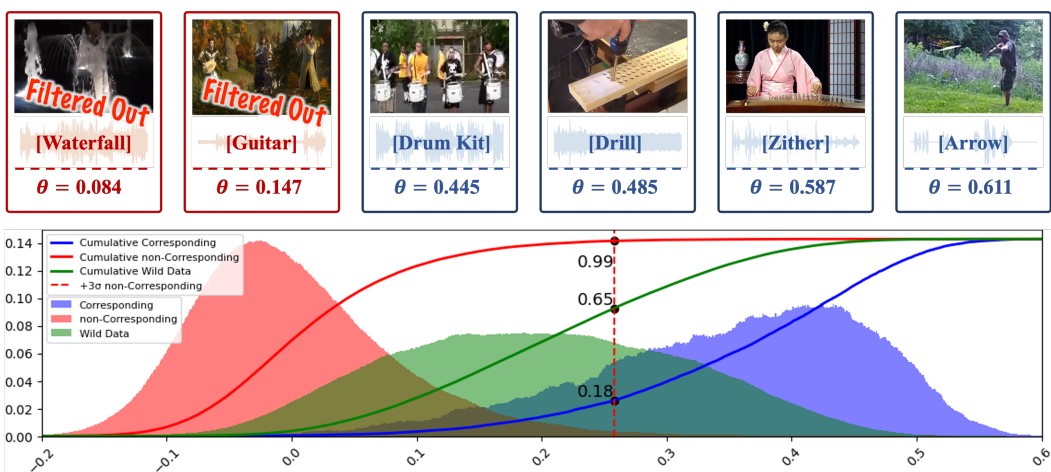

Figure 1: The distribution of audio-visual similarity among audio-visual corresponding samples, audio-visual non-corresponding samples and randomly selected wild samples. The similarity of non-corresponding data follows the distribution $N_{non-corresponding}(0.015, 0.081^2)$. Approximately 65% of the randomly selected wild samples and 18% of the audio-visual corresponding samples exhibit similarities below the $\mu + 3\sigma$ (0.2564) threshold of $N_{non-corresponding}$, suggesting a potential for these samples to be classified as audio-visual non-corresponding.

samples that demonstrate high audio-visual correspondence with reliable audio category labels to form AVSET-700K.

Additionally, to further expand the number of samples in each audio class, we select Panda-70M [8], a large-scale video-text dataset containing 70 million semantically consistent segments. It employs shot boundary detection technology [1] to divide the original videos into smaller semantically consistent segments. This segmentation ensures that each clip contains only a single event, preventing sound category conversion due to event switching and facilitating the subsequent filtering process. Leveraging Panda-70M, we expand AVSET-700K to a total of 10 million audio-visual corresponding samples, thus forming AVSET-10M.

**Stage 2: Audio-Visual Correspondence Filtering.** Previous researchers [6] compute the cosine similarity between textual class label and visual content to gauge alignment confidence between vision and language. They subsequently filter video samples for each class label using a manually selected threshold. However, this method is effective only in scenes featuring definite actions or visual objects. It struggles with abstract concepts, such as silence and urban outdoor scenes, even though these audios have specific associations with visual content. This consequently restricts the diversity of audio categories available in the dataset. We propose determining the confidence of audio-visual correspondence based on audio-visual similarity. This approach enables the screening of abstract audio samples and enhances the diversity of samples in the dataset.

Specifically, we randomly select 7,500 audio-visual corresponding samples $D_{corresponding}$ from the VGGSound dataset, and 7,500 wild data samples $D_{random}$ from the Panda-70M dataset. Additionally, we randomly construct 70,000 audio-visual non-corresponding samples $D_{non-corresponding}$ based on VGGSound. We employ Imagebind [13] to extract and calculate the cosine similarity between the average representation of 8 random video frames and the audio representation. The similarity distribution curves of different sample sets are depicted in Figure 1. The audio-visual non-corresponding samples exhibit a normal distribution $N_{non-corresponding}(0.015, 0.081^2)$, while random wild samples follow the distribution $N_{random}(0.211, 0.116^2)$. In contrast, the audio-visual corresponding samples exhibit a left-skewed distribution with a higher concentration of similar instances. When the similarity of samples exceeds the $\mu + 3\sigma$ (0.2564) threshold of the audio-visual non-corresponding distribution $N_{non-corresponding}$, they are considered audio-visual corresponding. Notably, only 35% of the randomly selected wild data samples exhibit similarities exceeding the $\mu + 3\sigma$ (0.2564) threshold of the distribution $N_{non-corresponding}$.

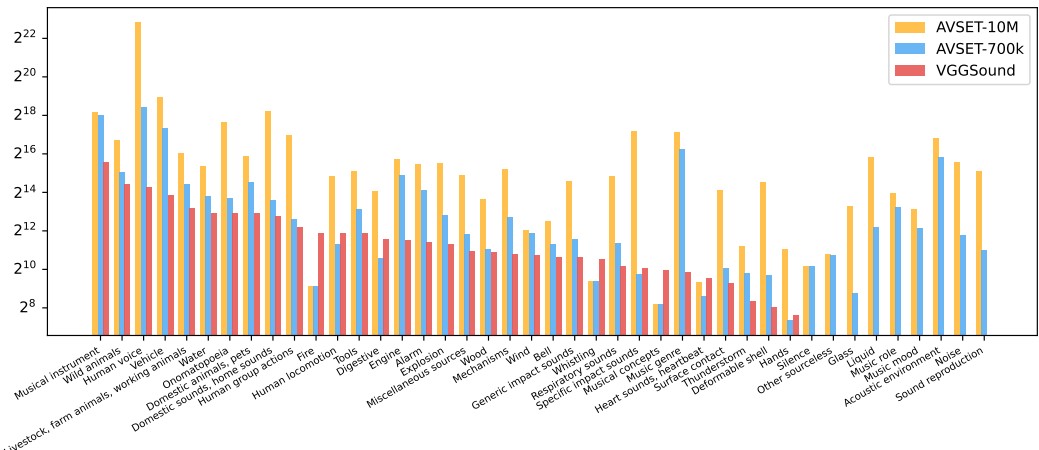

Figure 2: Comparison of the sample numbers for each audio category across AVSET-10M, AVSET-700K, and VGGSound datasets. Classification is carried out based on the secondary audio labels in AudioSet [3]. We pseudo-label each sample from Panda-70M using PANNs [21], while labels on VGGSound are manually aligned with AudioSet.

**Stage 3: Voice-Over Filtering.** While the aforementioned filtering method effectively identifies non-corresponding samples based on audio-visual similarities, it fails to account for samples containing background music and voice-overs. These off-screen sounds, largely irrelevant to the visual content, can disrupt the intended audio-visual correspondence. To address this issue, we utilize the audio classification network PANNs [21] to label each audio clip, specifically targeting and filtering out these voice-overs. Following the classification scheme used in AudioSet, we annotate each audio clip with seven primary audio categories and their respective sub-categories. Since speech and music are likely added during post-production, we specifically filter out samples that contain these elements along with other types of sounds. Other audio categories, such as the sounds of waterfalls and dog barking, typically originate from the original video. When these original video sounds co-occur with speech or music, it often indicates a high likelihood of off-screen voice interference. It is crucial to note that various instrumental sounds fall under the music category; thus, videos featuring instrumental performances are not excluded but are instead appropriately retained. Mirroring the approach in VGGSound [6], our filtering process aims to eliminate false positive samples—those with inappropriate sounds for each category. We refrain from using an audio classifier to select positive samples, as this may overlook some hard-to-classify yet criteria-meeting hard-positive audio samples.

**Stage 4: Sample Recycling with Sound Separation.** Speech is frequently encountered in wild video data, often leading to the inadvertent filtering out of a substantial amount of non-speech audio that includes voice-overs. This results in the loss of many potentially useful and valuable samples across various audio categories. Inspired by recent advancements in audio research [18], we have implemented a sound separation model[4] [33] that is specifically designed to isolate sounds that are neither speech nor music from audio mixes contaminated with voice-over noise. The outputs from this sound separation process are subsequently returned to Stage 2 to verify the correspondence between the newly isolated audio and the video.

## 3.2 Data Analysis

We perform comprehensive statistical analyses on the AVSET-10M and AVSET-700K datasets to gain detailed insights. For further information about these datasets, please refer to Appendix B.

**Diverse Categories, Abundant Samples.** Figure 2 presents a comparative analysis of the number of audio categories in AVSET-10M, AVSET-700K, and VGGSound. To ensure consistency in audio

---

[3]https://research.google.com/audioset/ontology/index.html
[4]https://github.com/ZFTurbo/MVSEP-CDX23-Cinematic-Sound-Demixing

Table 2: Comparison of sample numbers after each stage. Due to partial video corruption, we could only download part of the original dataset. [†] The numbers here represent the video clips we collected. AVSET-10M (w/o. AVSET-700K) represents samples filtered from Panda-70M.

| Stage | Goal | AVSET-700K | | AVSET-10M (w/o. AVSET-700K) | |
|---|---|---|---|---|---|
| | | #Num of Clips | Proportion | #Num of Clips | Proportion |
| $S1$ | Candidate Videos[†] | 1,445,360 | 100.0% | 39,295,551 | 100.0% |
| $S2$ | AV-C Filtering | 898,366 | 62.2% | 13,824,726 | 35.2% |
| $S3$ | Voiceover Filtering | 608,062 | 42.1% | 7,124,923 | 18.1% |
| $S4$ | Sample Recycling | 727,530 | 50.3% | 9,877,475 | 25.1% |

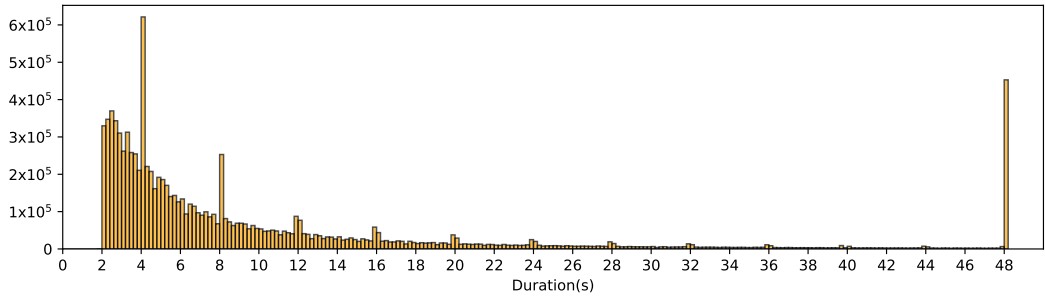

Figure 3: Histogram of Clip Length Distribution in AVSET-10M (w/o. AVSET-700K).

category labels across different datasets, we employ the PANNs [21] audio classification network trained on AudioSet to label all samples in AVSET-10M. Subsequently, we manually align the labels in VGGSound with those in AudioSet and standardized the audio labels across all three datasets as secondary labels. It is evident that AVSET-10M and AVSET-700K encompass a broader range of audio types compared to VGGSound, including categories such as silence, liquid, and glass. Furthermore, AVSET-10M significantly outperforms AVSET-700K and VGGSound in most categories, offering a greater number of audio samples for each audio category.

**Duration Statistics.** The samples filtered from Panda-70M include clips of varying lengths. As illustrated in Figure 3, we present the statistics for different clip lengths in AVSET-10M (excluding AVSET-700K). The total duration of AVSET-10M amounts to 30,418.6 hours, with an average clip length of 10.32 seconds. The longest clip spans 49 seconds, while the shortest measures 2 seconds. Notably, clips exceeding 10 seconds constitute 19,142.66 hours, representing 62.9% of total duration.

**The Number of Video Samples after Each Filtering Stage.** In Table 2, we detail the quantity of samples retained at each filtering stage for AVSET-700K and AVSET-10M (excluding AVSET-700K). Initially, in stage $S2$ for AVSET-10M (excluding AVSET-700K), we filter out 64.8% of video samples due to lack of audio-visual correspondence. In the subsequent $S3$ stage, 17.1% of the data containing voice-overs is removed. Further, in stage $S4$, an additional 8.0% of samples with voice-overs is refined through sound separation and subsequently recycled into the final audio-visual corresponding dataset. It is noteworthy that AudioSet undergoes a meticulous screening process by researchers, which results in a higher retention rate of data in the initial stage. AVSET-700K eliminates only 37.8% of data in its $S2$ stage.

### 3.3 Dataset Verification

We employ a distinct audio-visual representation learning model [41] different from the one used during the sample filtering phase to assess the reliability of our proposed sample filtering process. Specifically, we randomly sample data from four different audio-visual sources for validation: (1) audio-visual corresponding data from VGGSound, (2) audio-visual non-corresponding data created by randomly combining audio and video within VGGSound, (3) wild data randomly sampled from AudioSet, and (4) data from AVSET-700K obtained after the comprehensive filtering process. As

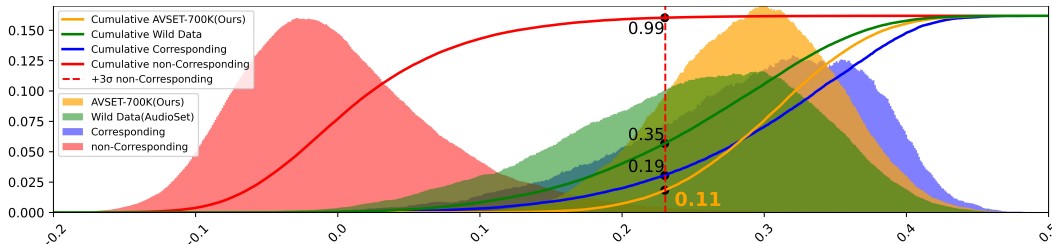

Figure 4: The distribution of audio-video cosine similarity of pre-trained model InternVL$_{IB}^{\dagger}$++(Ver.) [41] was evaluated on different sample sets: (1) the audio-visual ccorresponding samples from VGGSound, (2) the randomly combined audio-visual non- corresponding samples from VGGSound, (3) the wild samples from AudioSet, and (4) the AVSET-700K sample set filtered with complete dataset processing. Notably, only 11% of the samples in AVSET-700K fall below the $\mu + 3\sigma$ threshold of non-corresponding distribution $N_{non-corresponding}$.

depicted in Figure 4, we present the distributions of audio-visual similarity for these four sources. The mean and standard deviation of these similarities for each data source are detailed in Table 3.

**AVSET-700K vs. AudioSet.** It is evident that after data filtering, the audio-visual correspondence within the dataset is significantly enhanced compared to the wild data. The average cosine similarity of the AVSET-700K data increases from 0.258 to 0.303, while the standard deviation decreases from 0.086 to 0.058. Within the range $(\mu - 3\sigma, \mu + 3\sigma)$ of the normal distribution $N'_{non-corresponding}$ of non-corresponding data, the proportion of

Table 3: The mean and standard deviation (Std.) of audio-visual similarity among different sample sets.

| Sample Sets | Mean | Std. |
|---|---|---|
| Non-Corresponding (Random) | 0.015 | 0.072 |
| Wild Data (AudioSet) | 0.258 | 0.086 |
| Corresponding (VGGSound) | 0.302 | 0.083 |
| AVSET-700K (ours) | **0.303** | **0.058** |

potential non-corresponding samples is reduced from 35% to 11%. This improvement demonstrates that our sample filtering method effectively enhances the audio-visual correspondence in the dataset.

**AVSET-700K vs. VGGSound.** As an audio-visual corresponding dataset, VGGSound contains a large number of samples with high audio-visual similarity. However, a substantial portion of the data exhibits low similarity, with 19% of VGGSound samples falling below the $\mu + 3\sigma = 0.231$ threshold of the distribution $N'_{non-corresponding}$. In contrast, only about 11% of the samples in AVSET-700K have an audio-visual similarity below 0.231, indicating that AVSET-700K contains more samples with high audio-visual correspondence. Additionally, AVSET-700K features a smaller standard deviation and fewer samples exhibiting extremely low similarity, demonstrating that our sample filtering process effectively enhances the robustness of audio-visual correspondence.

## 4 Experiment

We benchmark two audio-visual tasks to explore the audio-visual correspondence: (1) Audio-Video Retrieval and (2) Vision-Queried Sound. In audio-video retrieval task, we experiment with AVSET-10M and focus on the data scale and the audio-visual temporally consistency. As for Vision-Queried Sound Separation, we mainly focus on the impact of each filtering stage, and work on the AVSET-700K which is of a similar scale to AudioSet. Specifically, we employ Imagebind [13] to extract the average features of 1 frame per second in the video as image features **I** and InternVid [40] to extract the features of the entire video as video features **V**. Please refer to Appendix A for additional details regarding the experiments.

### 4.1 Audio-Video Retrieval

For the audio-video retrieval task, we validate on two audio-visual corresponding datasets, AVE [36] and VGGSound [6], and compare the Recall@1 (R@1) and Recall@5 (R@5) from vision to audio. For the image+video (I+V) modality, we apply feature weighting similar to [41], with the mixed

Table 4: Comparison between the image-based method and the image+video based method on the task of visual to audio retrieval. The similarity on the diagonal should be the highest in each column. **The correct results** are highlighted in green, and **the incorrect results** are highlighted in red.

(a) Sample1 = $\{I_1,V_1,A_1\}$    (b) Sample3 = $\{I_3,V_3,A_3\}$

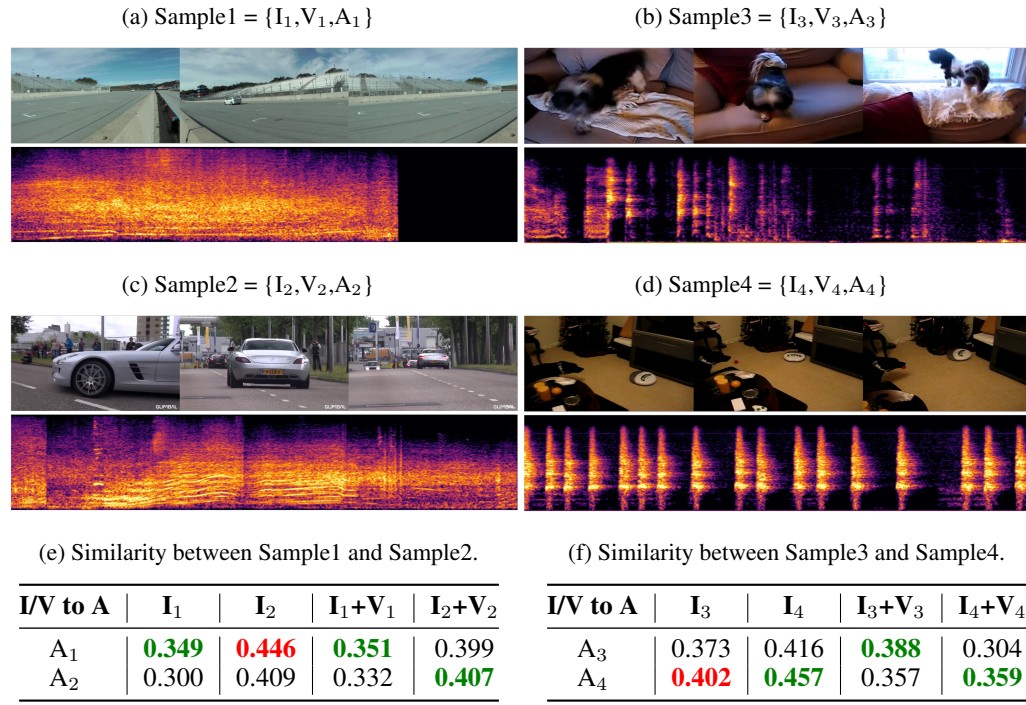

(c) Sample2 = $\{I_2,V_2,A_2\}$    (d) Sample4 = $\{I_4,V_4,A_4\}$

(e) Similarity between Sample1 and Sample2.

| I/V to A | $I_1$ | $I_2$ | $I_1+V_1$ | $I_2+V_2$ |
|---|---|---|---|---|
| $A_1$ | **0.349** | **0.446** | **0.351** | 0.399 |
| $A_2$ | 0.300 | 0.409 | 0.332 | **0.407** |

(f) Similarity between Sample3 and Sample4.

| I/V to A | $I_3$ | $I_4$ | $I_3+V_3$ | $I_4+V_4$ |
|---|---|---|---|---|
| $A_3$ | 0.373 | 0.416 | **0.388** | 0.304 |
| $A_4$ | **0.402** | **0.457** | 0.357 | **0.359** |

Table 5: Comparison of vision to audio retrieval performance using different methods on ASE and VGGSound. **M** denotes the visual features used during retrieval.

| #ID | M | Training Schedule | AVE | | VGGSound | |
|---|---|---|---|---|---|---|
| | | | R@1 | R@5 | R@1 | R@5 |
| $R1$ | I | AudioSet | 18.00 | 40.11 | 11.74 | 28.52 |
| $R2$ | I | AVSET-700K | 19.10 | 42.92 | 13.90 | 31.68 |
| $R3$ | I | AVSET-10M → AVSET-700K | 19.11 | 43.05 | 13.91 | 31.94 |
| $R4$ | I+V | AVSET-700K | 20.55 | 44.21 | 14.47 | 33.62 |
| $R5$ | I+V | AVSET-10M → AVSET-700K | **20.78** | **44.47** | **14.93** | **34.03** |

feature $f_{I+V}$ calculated as $f_{I+V} = 0.9f_I + 0.1f_V$. In all the audio-video retrieval experiments conducted for this paper, we train a separate linear layer for each modality to align representations across different modalities, using a batch size of 1024.

**AudioSet vs. AVSET-10M.** AudioSet contains a significant number of audio-visual samples that do not correspond, adversely affecting audio-video alignment. By employing our filtered dataset, AVSET-700, we enhance cross-modal alignment capabilities, achieving a 3.16% improvement in VGGSound R@5 performance from $R1$ to $R3$ in Table 5. Furthermore, expanding the dataset to 10 million ($R5$) entries boosts the model performance on AVE R@5 by an additional 0.26%.

**Based on Image vs. Based on Image+Video.** Previous models, which rely solely on image features to retrieve audio clips that semantically match the image, lake the capability to evaluate audio-visual temporal consistency. As shown in Table 5, by leveraging both image and video features, the R@5 performance on VGGSound improved by 2.09% from $R3$ to $R5$, emphasizing the importance of audio-visual temporal consistency.

**Qualitative Analysis.** Table 4 presents several qualitative results of audio-video retrieval, underscoring the importance of temporal consistency for effective audio-video retrieval. For example, the

Table 6: Comparison of sound separation performance among various methods on VGGSound. **M** stands for the query modality of sound separation.

| #ID | M | Training Schedule | VGGSound | |
|-----|---|-------------------|----------|---|
| | | | SDR↑ | SIR↑ |
| Baseline | | | | |
| $E1$ | I | VGGSound | 5.606±0.102 | 8.074±0.161 |
| $E2$ | V | VGGSound | **6.211±0.105** | **8.584±0.160** |
| Zero-Shot | | | | |
| $E3$ | V | AudioSet | 5.004±0.103 | 6.781±0.164 |
| $E4$ | V | AudioSet (w. AV-Correspondence Filtering) | 5.646±0.101 | 7.682±0.162 |
| $E5$ | V | AVSET-700K | **5.774±0.103** | **7.802±0.161** |
| Pretraining + Finetune | | | | |
| $E6$ | V | AudioSet (w. AV-Correspondence Filtering)→VGGSound | 6.548±0.103 | 9.251±0.158 |
| $E7$ | V | AVSET-700K→VGGSound | **6.666±0.102** | **9.377±0.158** |

image-based method could only deduce that engine roar should be present in the audio based on the image of a sports car, but it fails to determine when the sound should cease, leading to unsuccessful audio-video pairing. In contrast, when both image and video features are considered, the similarity between mismatched sample pairs 1 and 2 is reduced from 0.446 to 0.399, thereby achieving correct audio-video pairing.

## 4.2 Vision-Queried Sound Separation

As shown in Table 6, we present the performance of vision-queried sound separation based on different modalities across various datasets. We utilize the framework of CLIPSep [10] to implement sound separation models across various modalities.

**Image-Queried vs. Video-Queried.** Compared to the sound separation model based on image queries ($E1$), the model utilizing video queries ($E2$) demonstrates superior performance, with the Signal-to-Distortion Ratio (SDR) improving by 0.605. This enhancement highlights the importance of audio-visual temporal consistency within the audio-visual research.

**Corresponding vs. Non-Corresponding.** Audio-visual correspondence is critical for effective sound separation. Models trained on the non-corresponding AudioSet ($E3$) encounter difficulties in achieving accurate separation and fail to capture proper audio-visual alignment. After implementing audio-visual correspondence filtering ($E4$), the dataset shows a marked improvement in audio-visual correspondence, as evidenced by a 0.642 increase in the Signal-to-Distortion Ratio (SDR). Despite this advancement, the presence of voice-over content continues to challenge the alignment between audio and visual modalities. Following a comprehensive filtering process, the model ($E5$) trained on AVSET-700K exhibits exceptional zero-shot sound separation capabilities, achieving an SDR of 5.774. This significant enhancement underscores the effectiveness of our proposed filtering process.

## 5 Conclusion

Audio-visual correspondence datasets are vital for research in the audio-video field. Using a sophisticated sample filtering process with AudioSet and Panda-70M as sources, we develop AVSET-10M, the first open, large-scale dataset with high audio-visual correspondence, featuring ten million audio-visual corresponding samples across 527 audio categories. Verification experiments demonstrate that AVSET-10M surpasses previous datasets in terms of audio-visual correspondence. We also benchmark audio-video retrieval and vision-guided sound separation tasks, demonstrating the critical role of audio-video temporal consistency in this field. Our AVSET-10M provides greater opportunities for advancement in this field.

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

# A Implementation Details

## A.1 Sound Separation

Same as the experimental setting of [10], for all audio samples, we conduct experiments on samples of length 65535 (approximately 4 seconds) at a sampling rate of 16 kHz. For spectrum computation, we employ a short-time Fourier transform (STFT) with a filter length of 1024, a hop length of 256, and a window size of 1024. All images are resized to $224 \times 224$ pixels. All models are trained with a batch size of 128, using the Adam optimizer with parameters $\beta_1 = 0.9$, $\beta_2 = 0.999$, and $\epsilon = 10^{-8}$, for 200,000 steps. Additionally, we employ warm-up and gradient clipping strategies, following [10]. We compute the signal-to-distortion ratio (SDR) using museval [34]. All experiments are conducted on a single A800 GPU.

## A.2 Audio-Video Retrieval

Same as the experimental setting of [41], for all experiments, the softmax temperature is set to 0.01, and the temperature for the InfoNCE loss is set to 0.02. We utilize the Adam optimizer with a learning rate of $1 \times 10^{-3}$ and a batch size of 2048, running the training process for 20 epochs.

# B AVSET-10M

## B.1 Samples of AVSET-10M

We present some audio-video consistency samples from the AVSET-10M in Figure 5. For additional samples, please visit the demo page at `https://avset-10M.github.io`.

## B.2 Dataset Composition

We release AVSET-10M as the following two subsets:

- **AVSET-700K**: This subset comprises 727,530 audio-visual corresponding samples filtered from AudioSet. Each video segment in this subset is accompanied by a manually labeled audio category, ensuring accurate categorization and relevance.

- **AVSET-10M (w/o. AVSET-700K)**: This subset comprises 10,234,280 audio-visual corresponding samples, filtered from the Panda-70M dataset. Each video segment is semantically coherent, focusing on a single event, and includes a text description originally from the Panda70M dataset. Additionally, we provide pseudo-labels for the audio categories, derived with PANNs [21], along with their corresponding confidence scores. Researchers can use these pseudo-labels to freely partition the dataset.

We provide comprehensive meta-information for each video clip, including the URL of the video, timestamps for each clip, audio-visual cosine similarity, a flag indicating whether sound separation is required, and relevant text labels. For AVSET-10M (w/o. AVSET-700K), captions and pseudo-labels are included, while AVSET-700K features manual audio labels.

## B.3 Download URL

Please visit `https://avset-10M.github.io` to get the AVSET-10M. **Privacy Notice**: If any video clips in this dataset infringe upon your privacy, please contact us for their removal.

## B.4 LICENSE

AVSET-10M is released under the [CC BY 4.0] license. Before using this dataset, please ensure that you have read and understood the terms of the license.

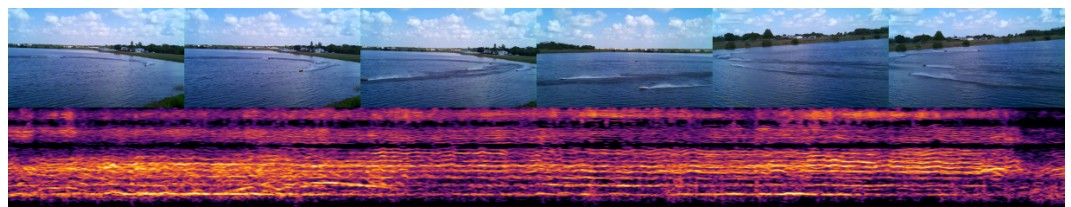

(a) Audio-Vision Cosine Similarity $\theta = 0.479$.

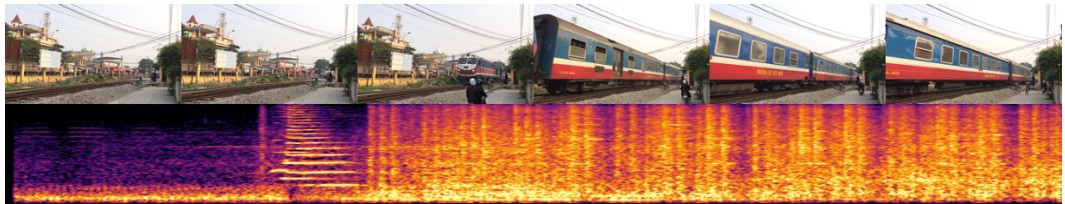

(b) Audio-Vision Cosine Similarity $\theta = 0.442$.

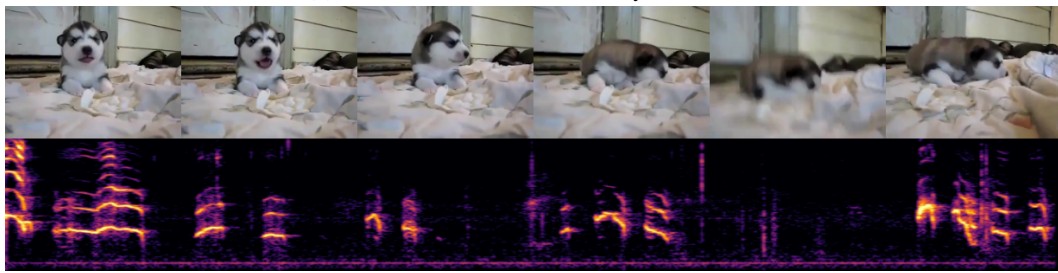

(c) Audio-Vision Cosine Similarity $\theta = 0.408$.

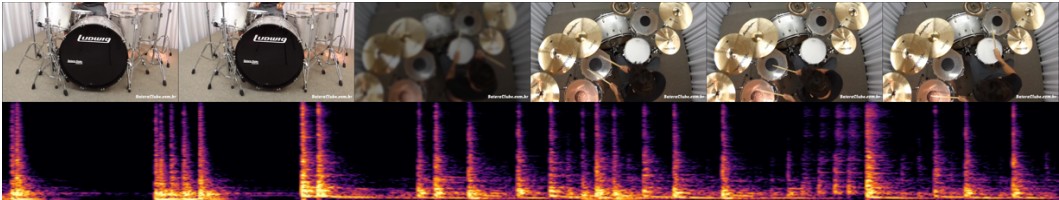

(d) Audio-Vision Cosine Similarity $\theta = 0.404$.

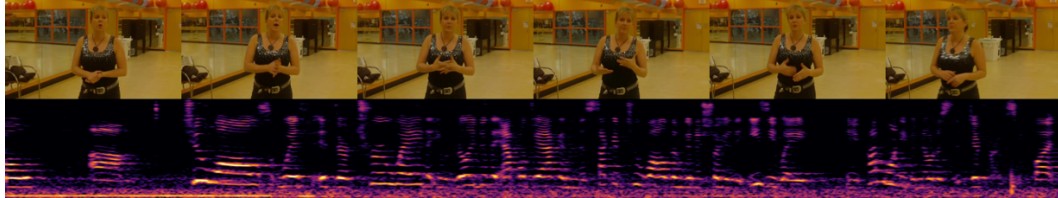

(e) Audio-Vision Cosine Similarity $\theta = 0.392$.

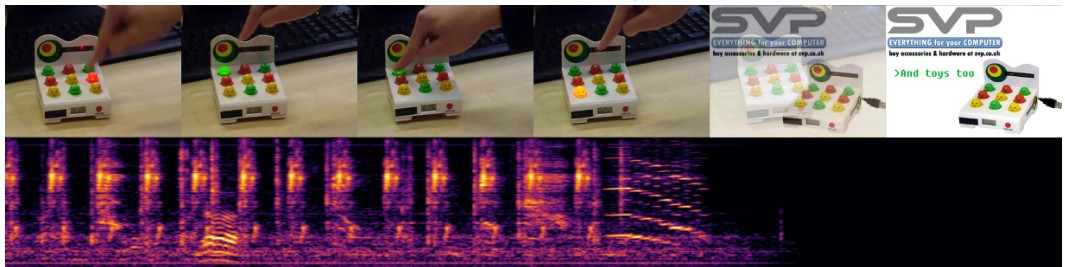

(f) Audio-Vision Cosine Similarity $\theta = 0.335$.

Figure 5: Audio-video consistency samples in AVSET.

## C Limitation

Since most existing video datasets predominantly contain clips with speech audio, which limits the amount of non-speech samples, we plan to utilize more diverse data sources in the future. This strategy aims to enhance the diversity of sample types and enable us to develop a more balanced and expansive dataset.

## D Ethical Impact

This paper primarily focuses on proposing a large-scale audio-visual correspondence dataset, aimed at expanding research possibilities in the audio-visual sector. This field includes technologies like video dubbing, which can lead to audio forgery. However, it's crucial to note that such dubbing does not result in severe identity forgery issues, unlike those caused by voice cloning technologies.

