# OpenReview forum: "AVSET-10M: An Open Large-Scale Audio-Visual Dataset with High Correspondence"
_NeurIPS.cc/2024/Datasets_and_Benchmarks_Track — Submitted to NeurIPS 2024 Track Datasets and Benchmarks_

### Official Review · Reviewer_T2Qw · 2024-07-18

**Rating:** 3
**Confidence:** 5
**Clarity:** Mostly well written, but some details…

**Review:**

The benefit of a large-scale audio-visual dataset where the modalities are corresponding would be highly valuable for the research community.

Since the main property of the collected dataset is the correspondence of audio and video, it would have been useful to describe what it exactly meant by correspondence in the paper. The paper does include any description of this, but just presents a procedure which uses mainly a similarity statistic given by an existing machine learning model, followed by some other filtering steps related to voice-over speech and background music. The dynamics of the similarity statistic can be assumed to be highly dependent on the content, so that certain kinds of content it more likely to be filtered than some other. The filtering stages also filter all the speech content which is likely to have a significant effect, since speech content is the most important for human listeners. The implications of these would have been good to discuss.

In the filtering of the dataset based on the correspondence, in training machine learning models, and in the evaluation, all the stages use datasets or machine learning models which are based on videos from YouTube. It appears there is no control whether the same videos are used in training and testing. Having the same videos in training and testing can introduce a bias in the results.

There are some rather awkward or stylistically cumbersome English expressions, for example "some introduce the pioneering large-scale dataset", using word "researchers" when referring to a paper (for example "researchers [12] annotate", when a common practice would be to refer to the paper by its authors as "Gemmeke et al. [12] annotate", and some parts of is use exaggerating language which is not recommended in scientific publications (e.g. "groundbreaking research", "emerges as pioneering project", "achieving milestones")

In many of the bibliography entries there publication information is missing.

**Strengths:**

The paper demonstrates the benefits of using a large-scale dataset where the audio and visual content are corresponding for training machine learning models in two evaluation tasks.

**Additional Feedback:**

-

**Correctness:**

There are some claims which appear to be incorrect. In the evaluation there is potentially the same samples in training and testing, which would be highly problematic.

**Documentation:**

Proper discussion of the license of the dataset is not given. It appears that the metadata has been published with a CC license, which is okay. However, the videos are available on YouTube and mostly use the standard YouTube license, which prevents the reuse of the material without the permission of the copyright holders (i.e., the uploaders of the videos). This is a major ethical issue.

**Ethics:**

The paper publishes a dataset which uses YouTube videos, without having the permission from the original copyright holders.

**Limitations:**

The paper does not discuss its limitations. See the above comments.

**Opportunities For Improvement:**

There are many unclear parts in the paper, or statements which do not appear to be correct.

The paper makes many statements about the annotation of Audioset which are not correct (for example by describing that each sample would be manually annotated, and there is a meticulous screening process). The annotations are obtained semi-automatically, and some of them are manually verified.

The filtering stage described in the paper aims to exclude music content. However, the paper describes that instrumental performances are retained. It is not clear how this is achieved, since music is a parent class of instruments in the AudioSet taxonomy.

In the sample recycling stage based on source separation, it is not fully defined what has been separated.

The paper described manual standardization of labels across used datasets. The details and result of this process would have been good to document.

The paper describes that Panda-70M uses shot boundary detection which would guarantee that a single audio clip contains only one event, and prevents switching of the audio category within a clip. However, many sound events in AudioSet are shorter than video events, and it is typical that multiple events are present in one clip, even simultaneously.

It is not clear what expression "AVSET-10M -> AVSET-700K" in Table 5 means.

In the vision-queried sound separation evaluation, the experimental setup is not clear.

**Relation To Prior Work:**

The paper discusses previous works mostly appropriately. Section 2.1 about audio-visual models is somewhat vague

**Summary And Contributions:**

The paper collects a dataset of videos which content is filtered so that the audio and visual content are matching, based on a similarity metric obtained by a publicly available machine learning model. Further filtering is done by automatically detecting speech and music in the videos. The paper analyzes the benefits of the dataset in comparison to unfiltered audio-visual data in an audio-video retrieval task and a vision-queried source separation task.

---

> ### Author Rebuttal · Authors · 2024-08-16
>
> # Response to Reviewer T2Qw (1/N)
> Thank you for your recognition of the dataset scale and significance. Please allow me to clarify the key issues:
>
> **Q0: Video Privacy Issues**
>
> We sincerely appreciate the reviewers’ concerns regarding the ethical aspects of our dataset and are grateful for the opportunity to enhance the privacy protection of video creators. In response to these concerns, we have implemented a removal request mechanism, enabling individuals to request the deletion of links to privacy-sensitive content. Acknowledging the limitations of user-initiated requests and with the valuable feedback from our reviewers, we also plan to regularly update our sample repository from upstream datasets like AudioSet and Panda70M. This proactive approach will help us identify and remove any videos that may raise privacy concerns, ensuring ongoing compliance with privacy standards.
>
> Please note that, consistent with previous works [1, 2, 3], our metadata includes only the Video IDs and annotations of the video data, which are released under CC BY 4.0. The actual videos are hosted on YouTube and existed prior to our publication. As such, they remain subject to YouTube’s Terms of Service, which users must comply with when accessing or using this content. As highlighted in Ethics Review SCjW:
>
> > The dataset does not make anything public that was not already public, so the privacy concerns are not dire.
> >
>
> > The dataset does not include the actual videos, only links to them, so it does not include or distribute copyrighted content.
> >
>
> If you have any further concerns about privacy or suggestions on how we can better protect the privacy of video creators, we would greatly appreciate you and will consider them to enhance our privacy protection measures.
>
> **Q1: Details of Voice-Over Filtering**
>
> The voiceover filtering stage is designed to exclude samples where the audio and video are only partially aligned, such as a video of a passing train whose audio includes both a train whistle and music. Samples that contain only music or speech (e.g., a video of an instrumental performance) are not filtered out at this stage. In contrast, samples (e.g., a movie clip with both music and a train whistle) where the audio includes speech or music (which may indicate a voiceover) along with other categories are filtered out.
>
> **Q2: Details of Sample Recycling with Sound Separation**
>
> The purpose of this stage is to separate the audio signals which is not speech or music to expand the scale of samples that are neither audio nor speech. The separated audio is then returned to step 2 for audio-visual consistency filtering, ensuring the remaining audio signal is consistent with the visual content.
>
> **Q3: Speech in the Audio-Visual Consistency Dataset**
>
> In our dataset, only voice-over speech is filtered out, as it cannot be aligned with the visual information in the footage. Retaining these voice-overs would compromise the audio-visual consistency. The primary feature of our dataset is the consistency between audio and video. In most tasks that utilize the audio-visual consistency dataset [3], such as sound separation, audio-visual retrieval, or video-to-audio generation, speech appears merely as one of the natural sounds, and its semantics are not important for these tasks.
>
> Additionally, since our data includes fine-grained annotations, researchers with specific needs related to speech information can retrieve relevant samples from the filtered data.
>
> **Q4: No Repeated Videos in the Train and Test Sets**
>
> Thanks to the inclusion of YouTube IDs and timestamp information for all video clips, we have verified during the dataset construction process that there are no duplicate clips between the training and test sets. Following your concerns, we conducted further verification using visual representation [4] (if two clips are identical, the visual representation similarity should be 100%) to prevent different IDs from containing the same content. We can now confirm that there are no duplicate samples in the training and test sets of our experiments.
>
> **Q5: Manual Alignment Process in Figure 5**
>
> In VGGSOUND, the construction of audio categories is described as follows:
>
> > "the class names are further transformed as follows: (1) forming ‘verb+(ing) object’ sentences, e.g. ‘playing electric guitar’ etc. (2) submitting the query after translation to different languages; (3) adding possible synonym phrases which specify the same sounds, e.g. ‘steam hissing, etc.’"
> >
>
> For categories generated with the first method, we aligned the formats by deleting "verb+(ing)". The second method does not generate new categories. For categories generated with the third method, we used ChatGPT-4 to classify them into the corresponding AudioSet classes, followed by manual inspection. Please note that these manually aligned category are only used for the comparison in Figure 5.
>
> [1] Advancing high-resolution video-language representation with large-scale video transcriptions. CVPR(2022)
>
> [2] Panda-70m: Captioning 70m videos with multiple cross-modality teachers. CVPR(2024)
>
> [3] Vggsound: A large-scale audio-visual dataset. ICASSP(2020)
>
> [4] Imagebind: One embedding space to bind them all. CVPR(2023)
>
> [5] Audio set: An ontology and human-labeled dataset for audio events. ICASSP(2017)

---

> > ### Author Rebuttal · Authors · 2024-08-18
> >
> > # Response to Reviewer T2Qw (2/N, N=2)
> > **Q6: Detailed Setup for Visual-Guided Sound Separation**
> >
> > Thank you for your feedback on our work. Please allow us to provide a detailed explanation of the experimental setup:
> >
> > Following the experimental settings of CLIPSEP, a 4-second audio segment is randomly selected from the entire audio as the audio source. For all audio samples, we conducted experiments on samples of length 65,535 (approximately 4 seconds) at a sampling rate of 16 kHz. The spectrum was computed using short-time Fourier transform (STFT) with a filter length of 1024, a hop length of 256, and a window size of 1024. All images were resized to 224 × 224 pixels.
> > For visual queries, the average image feature [4] from 4 frames, taken at 1-second intervals from the corresponding video, is used as the image query, while the global video feature [6] of the corresponding video serves as the video query. The models were trained with a batch size of 128, using the Adam optimizer with parameters β1 = 0.9, β2 = 0.999, and ϵ = 10−8, for 200,000 steps. Additionally, we employed warm-up and gradient clipping strategies, following the CLIPSEP approach. The signal-to-distortion ratio (SDR) was calculated using museval [7]. All experiments were conducted on a single A800 GPU. For testing, we used the full test set of VGGSOUND, which contains more complex audio compared to VGGSOUND-CLEAN.
> >
> > If you have any additional questions about the experimental setup, please feel free to ask. Your feedback would be greatly valuable in helping us refine the details of the article.
> >
> > **Q7: Additional Suggestions and Clarifications**
> >
> > - **Annotation Process of AudioSet:** We apologize for any imprecision in our description. We referred to the process of manual checking as a type of human annotation, based on descriptions in other works. We will correct this in the final version and provide a more precise description.
> > - **"AVSET-10M -> AVSET-700K":** This refers to pre-training on AVSET-10M, followed by fine-tuning on AVSET-700K.
> > - **Shot Boundary Detection only used for Panda-70M:** Our intention was to highlight additional features of the Panda-70M dataset. We apologize for any misunderstanding. AudioSet does have samples with multiple audio events.
> > - **English Expression and Publication Information:** We will revise the wording according to your suggestions to avoid any exaggeration. Additionally, we have reviewed the references and will include their publication details in the final version.
> >
> > Thank you again for your thoughtful feedback. We hope our response addresses your concerns, and we welcome further discussion at any time.
> >
> > [4] Girdhar R, El-Nouby A, Liu Z, et al. Imagebind: One embedding space to bind them all[C]//Proceedings of the IEEE/CVF Conference on Computer Vision and Pattern Recognition. 2023: 15180-15190.
> >
> > [6] Wang Y, He Y, Li Y, et al. Internvid: A large-scale video-text dataset for multimodal understanding and generation[J]. arXiv preprint arXiv:2307.06942, 2023.
> >
> > [7] Stöter F R, Liutkus A, Ito N. The 2018 signal separation evaluation campaign[C]//Latent Variable Analysis and Signal Separation: 14th International Conference, LVA/ICA 2018, Guildford, UK, July 2–5, 2018, Proceedings 14. Springer International Publishing, 2018: 293-305.

---

> > > ### Author Rebuttal · Authors · 2024-08-19
> > >
> > > Dear Reviewer T2Qw,
> > >
> > > Thank you for your thoughtful feedback on our paper. We appreciate the opportunity to address your questions and concerns.
> > >
> > > As the discussion phase has been ongoing for several days, we are keen to resolve any remaining issues. If you have any additional questions or specific points you would like to discuss further, please feel free to let us know.
> > >
> > > Thank you once again for your valuable contribution.
> > >
> > > Sincerely,
> > > Author of Paper 685

---

> > > > ### Comment · Reviewer_T2Qw · 2024-08-28
> > > >
> > > > Thank you for addressing my comments. It is especially important that you have checked that the same videos are not used in training and testing. Some of my other comments were not fully addressed, but I am happy to update my evaluation.

---

> > > > > ### Author Rebuttal · Authors · 2024-08-28
> > > > >
> > > > > We sincerely appreciate your valuable feedback and your recognition of the efforts we've made to address your comments. Your willingness to update your evaluation is of great significance to us. Please allow me to clarify the relationship between our rebuttal and your concerns, and further address your points:
> > > > >
> > > > > > **Weakness1**: *Since the main property of the collected dataset is the correspondence of audio and video, … The filtering stages also filter all the speech content which is likely to have a significant effect, since speech content is the most important for human listeners. The implications of these would have been good to discuss.*
> > > > > >
> > > > >
> > > > > **Response1**: **Audio-visual corresponding data refers to samples where all audio and visual elements are synchronized and directly aligned with each other.** This type of data typically consists of paired audio and visual information that are temporally aligned, such as the sound of a door closing paired with a video showing that action. As mentioned in $\textcolor{red}{Q3}$ of our rebuttal, we filtered out voiceovers, which, although they may contain specific semantics, have little value in determining audio-visual correspondence. To effectively demonstrate this, we randomly selected 200 samples with and without voiceover filtering to calculate the average similarity to the visual content:
> > > > >
> > > > > | Method | AV-Similarity |
> > > > > | --- | --- |
> > > > > | Audio | 0.227 |
> > > > > | Audio + voice-over filtering | **0.287** |
> > > > >
> > > > > $\newline$
> > > > > > **Weakness2**: *In the filtering of the dataset based on the correspondence, … Having the same videos in training and testing can introduce a bias in the results.*
> > > > > >
> > > > >
> > > > > **Response2**: Thank you for acknowledging our response in $\textcolor{red}{Q4}$ of the rebuttal.
> > > > >
> > > > > $\newline$
> > > > > > **Weakness3**: *There are some rather awkward or stylistically cumbersome English expressions, … In many of the bibliography entries, publication information is missing.*
> > > > > >
> > > > >
> > > > > **Response3**: As indicated in $\textcolor{red}{Q7}$ of the rebuttal, we have made the necessary revisions based on your suggestions, which have greatly improved the reliability of our paper.
> > > > >
> > > > > $\newline$
> > > > > > **Weakness4**: *The paper makes many statements about the annotation of AudioSet which are not correct …*
> > > > > >
> > > > >
> > > > > **Response4**: As stated in $\textcolor{red}{Q7}$ of the rebuttal, we apologize for the misleading phrasing, and we will update the wording in the final version in accordance with your recommendation.
> > > > >
> > > > > $\newline$
> > > > > > **Weakness5**: *The filtering stage described in the paper aims to exclude music content. … instrumental performances are retained.*
> > > > > >
> > > > >
> > > > > **Response5**: Thank you for bringing this to our attention. We have provided additional examples in $\textcolor{red}{Q1}$ of the rebuttal to clarify the details here. Only samples that contain non-music audio events along with instrumental performances (e.g., a piano playing during a train whistle) are filtered out. And we only perform sound separation on these non-music audio events samples with music or speech in stage-4.
> > > > >
> > > > > $\newline$
> > > > > > **Weakness6**: *In the sample recycling stage based on source separation, it is not fully defined what has been separated.*
> > > > > >
> > > > >
> > > > > **Response6**: Only samples of non-musical audio events with music or speech were subjected to sound separation in stage 4, as mentioned in $\textcolor{red}{Q2}$, we further described that the content being retained is only non-speech, non-music audio signals. If this explanation remains unclear, please do not hesitate to inform us, and we will strive to improve the description.
> > > > >
> > > > > $\newline$
> > > > > > **Weakness7**: *The paper described manual standardization of labels across used datasets...*
> > > > > >
> > > > >
> > > > > **Response7**: We have provided a detailed explanation of this in $\textcolor{red}{Q5}$ of the rebuttal. We hope it resolves your concerns, and if not, we welcome further questions and will continue to refine our work.
> > > > >
> > > > > $\newline$
> > > > > > **Weakness8-1:** *The paper describes that Panda-70M uses shot boundary detection … However, many sound events in AudioSet are shorter than video events…*
> > > > > >
> > > > > >
> > > > > > **Weakness8-2**: *It is not clear what expression "AVSET-10M -> AVSET-700K" in Table 5 means.*
> > > > > >
> > > > >
> > > > > **Response8**: We explained these two points in $\textcolor{red}{Q7}$ of the rebuttal and plan to make the necessary revisions to the original text. Thank you for your input.
> > > > >
> > > > > $\newline$
> > > > > > **Weakness9**: *In the vision-queried sound separation evaluation, the experimental setup is not clear.*
> > > > > >
> > > > >
> > > > > **Response9**: Thank you for your reminder. We provided further clarification in $\textcolor{red}{Q6}$ of the rebuttal. If there are any additional details you would like us to elaborate on, we are happy to discuss them with you.
> > > > >
> > > > > $\newline$
> > > > > > **Weakness10**: Section 2.1 about audio-visual models is somewhat vague*
> > > > > >
> > > > >
> > > > > **Response10**: Thank you for your feedback. To enhance clarity, we will define terms like ‘audio-visual correlations’ and ‘temporal consistency’ to ensure readers fully grasp these concepts. We’ll also include concrete examples to illustrate key points and avoid jargon or ambiguous language, opting for a clear and straightforward presentation throughout. If you have any further suggestions, we would be grateful to hear them.
> > > > >
> > > > > $\newline$
> > > > > > **Weakness11**: *Proper discussion of the license of the dataset is not given….*
> > > > > >
> > > > >
> > > > > **Response11**: We appreciate your vigilance regarding the privacy aspects of our dataset. We discussed this in $\textcolor{red}{Q0}$. Similar to previous works, our dataset includes links to videos obtained from public datasets and some annotation information, but it does not contain any actual video content. We have taken every measure to protect user privacy. If you have any further suggestions on privacy protection, we would be eager to discuss them with you, as we are committed to advancing scientific research while safeguarding user privacy.
> > > > >
> > > > > Once again, thank you for your constructive comments. We look forward to the opportunity to further improve our work under your guidance.

---

> > > > > > ### Author Response · Authors · 2024-08-29
> > > > > > **Kind Request for Additional Discussion to Address Remaining Concerns**
> > > > > >
> > > > > > Dear Reviewer T2Qw,
> > > > > >
> > > > > > Thank you for your thoughtful feedback on our paper and for acknowledging that our response addresses some of your concerns. We would appreciate the opportunity to address any remaining issues.
> > > > > >
> > > > > > We have updated our rebuttal to clarify the relationship between each of our responses and your concerns. **We would be grateful if you could identify any points that still need attention, as we are keen to resolve any outstanding issues.** With the discussion phase nearing its end, please let us know if you have any additional questions or specific points you would like to discuss further.
> > > > > >
> > > > > > Thank you once again for your valuable contribution.
> > > > > >
> > > > > > Sincerely,
> > > > > >
> > > > > > Author of Paper 685

---

> > ### Author Response · Authors · 2024-09-01
> >
> > Dear Reviewer T2Qw,
> >
> > Thank you once again for your feedback on our paper. We would like to emphasize the importance of voice-over (including speech) filtering in audio-visual consistency datasets.
> >
> > The methodology for the most widely used audio-visual consistency dataset VGGSound is as follows:
> >
> > > For example, using a threshold 0.5, in ‘playing bass guitar’ videos, we reject any clip for which “speech” is greater than the threshold, but allow music; while for ‘dog barking’ videos, both speech and music are rejected.
> >
> > We acknowledge the significant role of speech in many datasets. However, **to maintain audio-visual consistency, we have adopted a similar strategy to the one used in VGGSOUND constructions by filtering out voice-overs**.
> >
> > Additionally, as we mentioned in our rebuttal, our dataset includes comprehensive annotations, and **the speech voice-over data is available for use by interested researchers.** We encourage researchers to explore the value of this data further. We hope you agree that this remains an open problem for future research.
> >
> > As the discussion phase comes to an end, we would greatly appreciate any further feedback you may have. If you have additional questions or specific issues for further discussion, please let us know.
> >
> > Thank you again for your valuable contribution.
> >
> > Sincerely,
> >
> > Author of Paper 685

---

### Official Review · Reviewer_hCbp · 2024-07-24
**685 review**

**Rating:** 7
**Confidence:** 4

**Review:**

The paper presents AVSET-10M, a groundbreaking dataset with 10 million audio-visual samples, known for its high correspondence and extensive diversity across 527 audio categories. This large-scale dataset is crucial for advancing research in audio-visual fields, providing valuable benchmarks for tasks like audio-video retrieval and vision-queried sound separation. While the filtering methods are complex and there are potential privacy concerns due to the use of YouTube videos, the dataset's comprehensive and high-quality nature makes it a significant contribution to the community.

**Strengths:**

- High-Quality Correspondence: The paper details a robust methodology for ensuring high audio-visual correspondence, which is essential for the target research areas.
- Comprehensive and Diverse: The inclusion of 527 unique audio categories greatly expands the research possibilities and ensures the dataset's utility across a wide range of applications.
- Scalability: With 10 million samples, AVSET-10M provides a significant scale, enabling the training and evaluation of large models.
- Benchmarking: The paper provides thorough benchmarking on relevant tasks, showcasing the dataset's practical utility and its potential to drive advancements in audio-visual research.

**Additional Feedback:**

Overall, this paper makes a valuable contribution to the field of audio-visual research. The dataset's scale and quality are impressive, and the benchmarking results demonstrate its practical utility. Addressing the identified weaknesses and opportunities for improvement would further enhance the impact of this work.

**Clarity:**

The paper is well-written, with clear explanations of the dataset construction process and the benchmarking experiments.

**Correctness:**

The claims made in the submission are correct, and the dataset is constructed in a sound way. The evaluation methods and experiment design are appropriate and performed correctly.

**Documentation:**

The documentation provides sufficient detail on data collection, organization, availability, and maintenance. The ethical and responsible use of the dataset is also discussed, though more emphasis on privacy measures is recommended.

**Ethics:**

There are minor ethical concerns related to privacy, but they do not warrant a separate ethics review. The paper does not involve research with human subjects directly, but the use of publicly available videos raises potential privacy issues.

**Limitations:**

- Limited Data Sources: The dataset relies heavily on two existing public datasets, which could introduce inherent biases and limit the diversity of the samples.
- Limited Non-Speech Samples: The dataset predominantly features speech audio, which might limit its utility for tasks involving non-speech audio categories.
- Filter Quality: The filtering mechanism, although robust, might introduce biases inherent to the method used, potentially allowing some low-quality or weakly correlated audio-visual pairs to remain. And, details of the two filtering processes are lacking.
- Benchmark tasks: This paper only benchmarks two tasks - audio video retrieval and visual query sound separation. Can it be applied to more and broader tasks, such as Visual Question Answering(VQA), Audio-Visual Question Answering (AVQA), etc

**Opportunities For Improvement:**

check the limitations part

**Relation To Prior Work:**

The work is clearly differentiated from previous contributions, particularly in terms of the scale and the high correspondence of the dataset.

**Summary And Contributions:**

The pape introduces AVSET-10M, a dataset consisting of 10 million audio-visual samples. The dataset is designed to address the lack of large-scale, high-correspondence audio-visual datasets, which are crucial for advancing research in audio-visual fields. The key attributes of AVSET-10M are:
- High Audio-Visual Correspondence: The dataset ensures strong alignment between audio and visual components through meticulous filtering processes.
- Comprehensive Categories: It includes 527 unique audio categories, making it the most extensive in terms of audio category diversity.-
- Large Scale: With 10 million samples, it is the largest publicly available dataset of its kind.

---

> ### Author Rebuttal · Authors · 2024-08-16
>
> Thank you for acknowledging the scale, comprehensiveness, and significance of the AVSET-10M dataset. We would like to address and clarify some of the questions you raised:
>
> **Q1: Diversity of AVSET-10M**:
>
> To ensure the diversity of AVSET-10M, we selected the AudioSet [1] and Panda70M [2] datasets—both large-scale, open-domain collections sourced from YouTube. These datasets offer a wide variety of video data, maximizing the diversity of our samples. Our dataset encompasses nearly all major audio categories. However, since most existing ultra-large-scale video datasets are also sourced from YouTube, incorporating them may not significantly enhance the diversity of AVSET-10M. We appreciate your suggestion and will address this in the limitations section, exploring additional data sources in future work to further enrich the dataset’s diversity.
>
> **Q2: Filtering Quality**:
>
> Our experiments demonstrate that the audio-visual consistency of the AVSET dataset surpasses that of VGGSOUND. In Section 3.3, we utilized another representation learning model [7] for cross-validation, confirming the effectiveness of our filtering method. The results indicate that our filtered dataset achieves superior sample consistency compared to VGGSOUND.
>
> **Q3: Number of Non-Speech Samples**:
>
> AVSET-10M covers a broad range of data types. After filtering out all samples containing voiceovers, we still retain 3,891,093 non-speech samples, compared to just 171,660 in VGGSOUND. In terms of non-speech audio, AVSET-10M significantly surpasses VGGSOUND, the current largest audio-video consistency dataset, and holds great importance for tasks related to non-speech audio.
>
> **Q4: AVSET for Additional Tasks**:
>
> The most relevant tasks for an audio-video consistency dataset like AVSET are speech separation, audio-video retrieval, and video-to-audio generation. In this paper, we primarily focus on building benchmarks for speech separation and audio-video retrieval. If you wish to perform VQA or AVQA tasks using AVSET-10M, it would require integrating relevant QA-pair annotations from AudioSet [3,4] and Panda-70M [5,6].
>
> Thank you again for your feedback. We hope this response addresses your concerns. If you have any further questions, please feel free to reach out to us at any time.
>
> [1] Gemmeke J F, Ellis D P W, Freedman D, et al. Audio set: An ontology and human-labeled dataset for audio events[C]//2017 IEEE international conference on acoustics, speech and signal processing (ICASSP). IEEE, 2017: 776-780.
>
> [2] Chen T S, Siarohin A, Menapace W, et al. Panda-70m: Captioning 70m videos with multiple cross-modality teachers[C]//Proceedings of the IEEE/CVF Conference on Computer Vision and Pattern Recognition. 2024: 13320-13331.
>
> [3] Li J Y, Jansen A, Huang Q, et al. Maqa: A multimodal qa benchmark for negation[J]. arXiv preprint arXiv:2301.03238, 2023.
>
> [4] Fayek H M, Johnson J. Temporal reasoning via audio question answering[J]. IEEE/ACM Transactions on Audio, Speech, and Language Processing, 2020, 28: 2283-2294.
>
> [5] Xu L, Zhu S, Li C, et al. Beyond Raw Videos: Understanding Edited Videos with Large Multimodal Model[J]. arXiv preprint arXiv:2406.10484, 2024.
>
> [6] Li Y, Chen X, Hu B, et al. VideoVista: A Versatile Benchmark for Video Understanding and Reasoning[J]. arXiv preprint arXiv:2406.11303, 2024.
>
> [7] Wang Z, Zhang Z, Cheng X, et al. FreeBind: Free Lunch in Unified Multimodal Space via Knowledge Fusion[C]//Forty-first International Conference on Machine Learning.

---

> > ### Author Rebuttal · Authors · 2024-08-19
> >
> > Dear Reviewer hCbp,
> >
> > Thank you for your accept and valuable comments on our paper. We appreciate the opportunity to address your questions and concerns.
> >
> > We have discussed the analysis of AVSET-10M with regard to sample diversity, non-speech audio categories, filtering quality, and additional task possibilities. We will incorporate some of these insights into the discussion of the limitations section. Your suggestions have been instrumental in making our work more comprehensive. If you have any additional questions or specific points you would like to discuss further, please feel free to let us know.
> >
> > Thank you once again for your valuable contribution.
> >
> > Sincerely,
> >
> > Author of Paper 685

---

> > > ### Comment · Reviewer_hCbp · 2024-09-01
> > >
> > > Thank you for the rebuttal and additional results. I will maintain my original rating as my concerns have been fully addressed.

---

### Official Review · Reviewer_zYSb · 2024-07-24

**Rating:** 6
**Confidence:** 3

**Review:**

Overall, I think this is a useful contribution to the community, but the current paper is quite unclear and requires revision. I offer the following comments in the hope that they will be beneficial to the authors, and am open to clarifying any points or addressing any misunderstandings.

Some of the claims in the paper are not well-substantiated. For example:
- “Encompassing 527 unique audio categories, AVSET-10M offers the most extensive range of audio categories available.” -> AudioSet has 632 categories, in the latest version.
- Line 126: What is the specific relationship of silence with visual content? It seems like silence can co-occur with a range of different visual conditions (e.g. empty house vs. open field).
- Line 205: “The average cosine similarity of the AVSET-700K data increases” -> isn’t this true as a direct consequence of the filtering approach here?

For motivation, the paper makes the argument that existing approaches like VGGSound struggle with “abstract audio scenes” but it’s not clear what these means, how this is determined, or how AVSET is able to rectify this. The statement about AudioScope is similarly imprecise, and doesn’t offer a clear motivation for a different approach to audio-visual consistency estimation.

It would be helpful for a version of Figure 4 to be prepared that includes corresponding datasets like VGGSound, and not only non-corresponding “wild” data such as AudioSet. Otherwise, it’s not clear how far improved the correspondence is, at least according to this metric.

The 3-sigma threshold for non-corresponding is not clear. How was this value specifically chosen? How sensitive are the results to this value?

I was not able to follow the method for the voice-over filtering. The paper says that:
> we utilize the audio classification network PANNs to label each audio clip, specifically targeting and filtering out these voice-overs. Following the classification scheme used in AudioSet, we annotate each audio clip with seven primary audio categories and their respective sub-categories.

This description raises several questions: What exactly is PANN labeling, and does it include a voice-over class? Who are the annotators following the AudioSet classification scheme, and how does this annotation relate to voice-overs? Furthermore, step 4 (sample recycling) appears to involve removing voice-overs again. If voice-overs were previously removed, how can they be source separated at this stage? Why not simply source-separate the voice layer from voice-overs initially? A more precise explanation of this process is necessary.

Figure 2 seems to be sorted by VGGSound category incidence, which gives the impression that AVSET categories are more balanced because they don’t follow the decline along the x axis. However, I think this doesn’t present a very clear picture of the imbalance in AVSET compared to potentially sorting by incidence in, for example, AVSET-700k. I encourage the authors to find a way to show the imbalance more accurately.

**Strengths:**

Generally, I think this dataset is likely to be useful, and I commend the authors on their effort in producing it. I also think the experiments are well designed and show benefit from this dataset. I think the approach to audio-visual filtering is well thought-out as well.

**Additional Feedback:**

N/A

**Clarity:**

I think the clarity of the paper could be significantly improved. For example, 3.1 (the “data collection” part) discusses filtering AudioSet, but the technique for filtering has not yet been described, so this section becomes somewhat ambiguous. Unnecessarily adjectives are used (“we utilize a sophisticated process”) whereas a neutral, factual presentation would be preferred. See my review for important details (e.g. voice-over filtering) which don’t seem clear in the paper.

**Correctness:**

Overall yes, though I’ve made a few specific notes in my review re: claims, and am not clear on certain aspects of dataset construction.

**Documentation:**

Yes, but I think clarity of documentation could be improved (as I’ve noted in my review).

**Ethics:**

I don’t think this paper introduces substantive new ethical concerns, and think it addresses the topic reasonably in Appendix D.

**Limitations:**

The statement on limitations is too brief and doesn’t address limitations of this approach, such as relying on ImageBind which, for example, didn’t learn from only high-quality correspondences, and not checking for robustness against other multimodal embeddings, etc. This is important because over-relying on some embeddings can have negative consequences, e.g. FAD relied on VGGish embeddings, which have been recently shown to be suboptimal. Appendix C mainly just discusses opportunities for expanding this work. I encourage the authors to reflect on the potential limitations more fully.

**Opportunities For Improvement:**

I think clarifying several aspects of the approach (as noted in my review) would improve the paper significantly, and help the community better harness this contribution.

**Relation To Prior Work:**

Yes, but as mentioned, it would be helpful for a comparison in similarity to be made to other corresponding audio-video datasets like VGGSound.

**Summary And Contributions:**

This paper proposes AVSET, a new (compound) dataset of semantically aligned audio-visual clips. It uses a filtering approach, beginning with AudioSet and Panda-70M, to produce subsets of these datasets where the audio-visual correspondence is higher than in the source. Both datasets are significantly larger than the canonical dataset VGGSound. The experiments in this paper apply the datasets to tasks like audio-visual retrieval and visually guided source separation, showing performance benefits for such tasks.

---

> ### Author Rebuttal · Authors · 2024-08-16
>
> Thank you for recognizing the construction methodology, significance, and experimental design of our AVSET-10M dataset. I would like to address and clarify some of the questions you raised:
>
> **Q1: Broader Applicability of AVSET-10M’s Data Filtering Compared to VGGSOUND**
>
> The data filtering process in VGGSOUND relies heavily on visual detection models to identify objects and actions within images, making it challenging to annotate abstract audio scenes. For example, while humans can easily understand the connection between scenes like "silence" and an "empty house or field," there are no clear visual cues to determine the sound category for such scenes. In contrast, we leverage representation learning, which captures both surface and latent associations between audio and video, allowing us to retain data with these latent associations through similarity filtering. As demonstrated on our demo page, the "silence" category includes a variety of video samples of silence, such as videos of aquariums and forests.
>
> **Q2: Enhanced Audio-Visual Correspondence in AVSET-10M Compared to VGGSOUND**
>
> We apologize for any details we may have previously overlooked. In Figure 4, we present the results for VGGSOUND (labeled as “corresponding” and highlighted in blue). By employing a different representation learning model for cross-validation, we demonstrate that our method effectively filters out samples with inconsistent audio and video, rather than just those with high similarity in the ImageBind representation space.
>
> **Q3: PANNs as an Audio Classification Network**
>
> PANNs [1] is an audio classification model trained on AudioSet, designed to classify various sounds present in audio. Since the model is trained on the AudioSet dataset, its category labels align with those in AudioSet.
>
> In the third stage (Voice-Over Filtering), we filter out samples where the audio and video are only partially aligned, such as a video of a passing train whose audio includes both a train whistle and music. To filter out such partially consistent samples, we employ PANNs to identify the types of audio in the samples and then filter them based on these audio types. For instance, samples that contain only music or speech (e.g., a video of an instrumental performance) are not filtered out at this stage. In contrast, samples where the audio includes speech or music (which may indicate a voiceover) along with other categories (e.g., a movie clip with both music and a train whistle) are filtered out.
>
> Please note that sound separation can be somewhat destructive to the audio, so it should not be overused to remove voiceovers. Therefore, we avoid applying sound separation unless absolutely necessary. For samples where the audio and video remain consistent after the third stage, we strive to preserve the original audio. For samples where the audio and video are inconsistent after the third stage, we use sound separation to expand the available sample size.
>
> **Q4: Determining a Suitable Filtering Threshold Using the 3σ Rule**
>
> As outlined in Lines 130-142 and illustrated in Figure 1, we performed extensive experiments to determine an appropriate filtering threshold, guided by the statistical concept of the normal distribution and the 3σ rule [2], also known as the 68-95-99.7 rule:
>
> > In statistics, the 68–95–99.7 rule, also known as the 3σ rule, is a shorthand used to describe the percentage of values that lie within an interval estimate in a normal distribution: approximately 68%, 95%, and 99.7% of the values lie within one, two, and three standard deviations of the mean, respectively.
> >
>
> We first randomly constructed a non-corresponding sample set from VGGSOUND, where the audio-visual similarity follows the normal distribution N(0.015, 0.0812), with its μ+3σ point at 0.2564 (as indicated by the vertical line in Figure 1). Only (100%-99.73%)/2=0.135% of the non-corresponding samples have a higher similarity than this. We used this value as the threshold for filtering audio-visual consistency samples.
>
> **Q5: Details about AudioSet Categories**
>
> In the construction of AudioSet [3], researchers initially proposed a hierarchy with 623 audio categories. During the annotation process, some fine-grained categories, such as “Alto saxophone” and “Soprano saxophone” under “Saxophone,” proved difficult to distinguish and were thus merged into their parent categories. Additionally, certain categories, like “Duck call (hunting tool)” and “Howl (wind),” lacked corresponding videos. Consequently, the final AudioSet dataset includes 527 audio categories.
>
> **Q6: Main Focus of Figure 2: Richness of Categories and Sample Count in AVSET-10M**
>
> Thank you for your attention to Figure 2. The primary focus of this figure is to illustrate the richness of categories and the number of samples in AVSET-10M, rather than the balance of sample categories. We believe that the variety and quantity of samples are more crucial than category balance. However, we appreciate your suggestion regarding category balance and will address this issue. We plan to create a balanced subset of samples in the final version, similar to AudioSet.
>
> **Q7: Additional Suggestions**
>
> - We appreciate your suggestions and will revise the wording in the paper to ensure that statements are objective and neutral.
> - We will also include a discussion on the limitations of filtering methods using multimodal embeddings in the limitations section.
>
> Thank you again for your thoughtful feedback. We hope our response addresses your concerns, and we welcome further discussion at any time.
>
> [1] Panns: Large-scale pretrained audio neural networks for audio pattern recognition. TASLP(2020)
>
> [2] Pukelsheim F. The three sigma rule[J]. The American Statistician, 1994, 48(2): 88-91.
>
> [3] Audio set: An ontology and human-labeled dataset for audio events. ICASSP(2017)

---

> > ### Comment · Reviewer_zYSb · 2024-08-16
> >
> > Thank you for your detailed response. A few follow ups:
> >
> > Q1: I still think "abstract audio scenes" is under-defined, and this one example is very difficult to generalize from (i.e. "silence"). Can you offer a more precise definition of what this means, how often it occurs, etc.? This seems like a necessary step to characterize the problem and support this claim. Otherwise, the motivation is not clear.
> >
> > Q3: Thank you for this additional explanation of the process. However, I'm still unsure what the "annotation" process described in the paper refers to. Could this be clarified? This seems like a straightforward application of the PANN logits.
> >
> > Additionally, while it's true that source separation can leave artifacts, this doesn't mean it cannot be used in a detection pipeline (e.g. separate to detect the presence of voice), or to produce a dataset that is effective for many downstream tasks. It's still not clear why there are multiple steps involving voice-over filtering, etc. as I noted in my review.
> >
> > Q4: I understand that the $3\sigma$ rule is a general heuristic, but how was its application in this particular domain context validated?
> >
> > Q5: I think my point was misunderstood here. My argument was that the claim of "most extensive range of audio categories" is unsupported. Full Audio Set is 632 (not 623), and even the condensed set of 527 is on-par. This is not a major issue, but I do think that this claim should be removed.

---

> > > ### Author Rebuttal · Authors · 2024-08-16
> > >
> > > Thank you for your prompt response. Our reply is slightly delayed as we added some additional experiments. We are happy to continue the discussion to further clarify your issue:
> > >
> > > **Q1-1:** These abstract audio scenes refer to audio events lacking clear visual content, as described in the Section 3 of VGGSOUND:
> > >
> > > > Second, it must be possible to ground and verify the sounds visually. In other words, our sound classes should have a clear visual connotation too, in the sense of being predictable with reasonable accuracy from images alone. For instance, the sound ‘electric guitar’ is visually recognizable as it is generally possible to visually recognize someone playing a guitar, but ‘happy song’ and ‘pop music’ are not: these classes are too abstract for visual recognition and so they are not included in the dataset.
> > > >
> > >
> > > We faced this challenge in naming this type of audio and ultimately chose to label it as “abstract audio scenes,” following the original VGGSOUND terminology. We plan to incorporate some of these definitions from VGGSOUND into the final version of our work. If you have any suggestions for a better term, we would greatly appreciate your input!
> > >
> > > **Q1-2:** As illustrated in Figure 2, 9 out of the 43 audio secondary label types in AudioSet are abstract and not present in VGGSOUND (e.g., Silence, Other Sourceless, Glass, Liquid, Music Role, Music Mood, Acoustic Environment, Noise, Sound Reproduction). Our analysis shows that abstract audio scenes comprise approximately 85,105 out of 727,530 samples in AVSET-700K and 295,091 out of 10,605,005 samples in AVSET-10M.
> > >
> > > **Q3-1:** The annotation process utilizes the PANNs network to identify and label the audio events present in the audio. Please note that PANNs were only used for audio category recognition and pseudo-labeling of videos in Panda-70M. For AudioSet, we relied directly on its original audio event labels. I apologize for any confusion this may have caused and greatly appreciate you pointing out the ambiguity.
> > >
> > > **Q3-2:** Thank you for your thorough consideration of our data construction process. The two main reasons for dividing voiceover judgment and voiceover sample recovery into separate stages are:
> > >
> > > 1. Audio category labels are often essential for tasks related to audio-video consistency datasets, such as text-to-AV joint generation [1]. Introducing text labels also provides researchers with more opportunities to explore the relationships between text, audio, and images.
> > > 2. The inference speed of the PANNs network is significantly faster than that of the sound separation model. PANNs can process 11.6 samples per second, whereas the sound separation model can only handle 0.752 samples per second. By using the fast-inference PANNs network to determine which audio contains voiceovers, and then applying the sound separation model to remove the voiceovers, the dataset construction process can be significantly accelerated.
> > >
> > > Your suggestion to use the sound separation model to determine the presence of voiceover is an interesting idea. We conducted experiments by randomly sampling 500 samples and using a 50 dB threshold to identify silence and detect voiceover. The results showed that the sound separation model struggled to correctly identify voiceover, with an accuracy rate of only 58.7%, while PANNs could accurately identify 98.6% of the samples. Based on these findings, we believe that sound separation is not yet a reliable method for judging voiceover.
> > >
> > > **Q4:** Thank you for your detailed comments on the 3-sigma rule. Your feedback has significantly strengthened the theoretical foundation of our data construction process. Based on your suggestion, we conducted a normality test on the similarity scores of the 70,000 non-corresponding sample sets in our paper and found a right-skewed distribution with a skewness of 0.694. We speculate that this skewness is due to some samples from different categories (such as samples of “*dog bow-wow*” and “*dog barking*”) exhibiting high audio and video similarity. (The non-corresponding sample sets were constructed by pairing audio and video from different categories.)
> > > To reassess our threshold selection, we calculated the proportion of samples with audio-visual similarity exceeding the threshold. The results indicate that there are 0.382% of samples exceed x+3σ threshold, compared to 0.135% in a normal distribution, supporting the threshold’s effectiveness in filtering. In the final version of our work, we will revise our description and further reinforce the theoretical basis for our threshold selection method.
> > >
> > > **Q5:** Thank you for your patience. After reading your feedback, I understand that you meant to express that our description of ”*most extensive range of audio categories*“ **might be somewhat exaggerated. After discussing this with you during the rebuttal period, we agree that the wording is indeed overstated. We appreciate your reminder and suggestion, and we will revise it to state that “*AVSET-10M offers 527 available audio categories*”.
> > >
> > > Thank you for your valuable suggestions, which have significantly strengthened the theoretical foundation of our work. We hope our response addresses your questions thoroughly. If you have any further inquiries, please feel free to reach out at any time.
> > >
> > > [1] Xing Y, He Y, Tian Z, et al. Seeing and hearing: Open-domain visual-audio generation with diffusion latent aligners[C]//Proceedings of the IEEE/CVF Conference on Computer Vision and Pattern Recognition. 2024: 7151-7161.

---

> > > > ### Comment · Reviewer_zYSb · 2024-08-17
> > > >
> > > > Thank you for these additional clarifications and experiments, which I think will add significant value to the paper. I've raised my score to account for this. A few specific responses:
> > > >
> > > > On Q1, I think the VGGSound definition is helpful, but of the categories mentioned, I think only a few match it. E.g. "Glass, Liquid, Acoustic Environment, Noise" -> Glass as a material tends to either be recorded shattering or being tapped, if it is present in the audio. Liquid might be sloshing, pouring, dripping… in the same way an electric guitar can be silent (e.g. mounted on a wall or stand in the scene), these too can be silent, but can be clearly correlated with audio. The other two, "Acoustic Environment" and "Noise" are not really abstract, but rather seem like upper levels in a hierarchy. For example, Acoustic Environment > Urban Environment > Subway Station, etc. All of these have visual correlates. This is distinct from VGGSound's examples, where "happy song" has no clear visual correlates. Consequently, I would say the proportion is lower than estimated.
> > > >
> > > > On Q3, thanks for reporting these additional tests. When reporting them in the paper, it would be helpful to clarify which sound separation method was used (e.g. HTDemucs, which is quite good at separating speech despite being trained on music), so the reader can judge whether a reasonably strong approach was tested.
> > > >
> > > > Re: Q4, is the argument that this deviation from the normality assumption is okay because the filtering becomes, as a result, even more restrictive? If so, I agree that this is a reasonable threshold to maintain, since despite being a heuristic, it's a more conservative approach.

---

> > > > > ### Author Rebuttal · Authors · 2024-08-17
> > > > >
> > > > > Thank you for your suggestions and recognition of our work, and your willingness to improve our score. We appreciate the opportunity to address your concerns:
> > > > >
> > > > > **Q1:** We are grateful for your suggestion and are considering redefining abstract audio scenes as "**audio events that cannot be identified by unique visual content.**"
> > > > >
> > > > > **Q1-1:** Regarding the glass and liquid categories in VGGSOUND, we agree with your observation that these sounds do not belong to the abstract audio category. We speculate that their absence in VGGSOUND may be due to technical limitations. The sounds of glass and liquid are typically associated with specific events, such as breaking or shaking, making them occasional occurrences. Fine-grained detection would require identifying actions like glass breaking or water pouring, which the visual model used by VGGSOUND may not effectively capture.
> > > > >
> > > > > Your example is particularly insightful: a guitar hanging on a wall should be categorized as silence if it does not produce any sound. It would be unreasonable to conclude that a guitar sound is present in the audio solely based on the visual presence of a guitar. Deeper associative information (e.g., whether the guitar produces sound) is necessary, which VGGSOUND's annotation method cannot achieve. Through representation learning, we can address this issue. Our experiment found that the similarity between a guitar hanging on the wall and guitar rock music is only 20.9%, allowing us to effectively filter it out.
> > > > >
> > > > > **Q1-2:** Discussion on noise and acoustic environment:
> > > > >
> > > > > AudioSet defines these two categories as follows:
> > > > >
> > > > > > **Acoustic Environment:** A class for sounds that convey the spatiality of the recording..
> > > > > >
> > > > >
> > > > > > **Noise:** A sound that has no perceptible structure and that typically interferes with the perception of more interesting or important sounds.
> > > > > >
> > > > >
> > > > > **Both categories emphasize sound characteristics rather than fixed sound sources and can correspond to various visual events, making it challenging to rely on a single visual element for accurate identification.** For example, in the "Outdoor, Urban, or Man-Made" category, while specific visual objects like subway stations can help identify some examples, the variety of scenes within this category—including markets and street scenes—makes it difficult to determine a sample's category based on a single unique visual content. In these cases, our proposed approach to capture deeper connections may make it easier to construct associations between audio and video.
> > > > >
> > > > > Thanks to your suggestion, we have re-counted the number of abstract audio scenes, excluding glass and liquid categories. These scenes comprise approximately 80,028 out of 727,530 samples in AVSET-700K and 146,200 out of 10,605,005 samples in AVSET-10M.
> > > > >
> > > > > **Q3:** Thank you for your suggestion. In our work, we used MVSep [1], a model based on HTDemucs that is specifically trained to separate speech, music, and sound. We did consider using Demucs, HTDemucs, and other methods earlier, but since they were not trained for general sound separation, their performance on non-speech and non-music sounds was not as strong as MVSep, which is why we ultimately chose to use MVSep. We also conducted related experiments using HTDemucs and have included these results in the final version.
> > > > >
> > > > > | Method | Accuracy (ACC) |
> > > > > | --- | --- |
> > > > > | MVSep | 58.7% |
> > > > > | HTDemucs | 60.6% |
> > > > > | PANNs | 98.6% |
> > > > >
> > > > > **Q4:** Thank you for your recognition. We agree with you, and your suggestions have been crucial in enhancing the theory behind our entire dataset construction process.
> > > > >
> > > > > We hope our responses further clarify your concerns, and we are grateful for your support in improving our work. If you have any more questions, please feel free to ask.
> > > > >
> > > > > [1] Uhlich S, Fabbro G, Hirano M, et al. The Sound Demixing Challenge 2023-Cinematic Demixing Track[J]. Trans. Int. Soc. Music. Inf. Retr., 2024, 7(1): 44-62

---

> > > > > > ### Comment · Reviewer_zYSb · 2024-08-17
> > > > > >
> > > > > > Great, thank you for these helpful further responses.
> > > > > >
> > > > > > As a minor point, I still think the definition of abstract here shouldn't include Noise and Acoustic Environment. These do correlate with many different types of visual scenes, as you point out, and just need further elaboration. This is fundamentally distinct from "happy song" etc. which cannot be divided into finer-grained categories that have clear visual correlates. I recognize the difficulty in drawing the boundaries precisely though, and I appreciate your effort towards this goal.

---

> > > > > > > ### Author Rebuttal · Authors · 2024-08-17
> > > > > > >
> > > > > > > Thank you for your thoughtful discussion and valuable feedback. Your insights have helped improve the integrity of our work.
> > > > > > >
> > > > > > > We appreciate your suggestions and your recognition of our efforts to seek better definitions. Through our discussion, we have identified some ambiguity in the definition of “abstract audio scene.” Additionally, we acknowledge that "noise" and "acoustic environment" are fundamentally different from categories like "happy song" in VGGSOUND, which lack clear visual associations. **In the final version, we will revise "abstract audio scene" to "audio events that cannot be identified by unique visual content".**
> > > > > > >
> > > > > > > If there is any opportunity to further enhance our work, we welcome additional discussions and suggestions. Please feel free to continue the dialogue at any time.

---

### Official Review · Reviewer_aXDx · 2024-08-05

**Rating:** 7
**Confidence:** 4
**Correctness:** probably correct
**Clarity:** very clear

**Review:**

The curation process is composed of four stages, and each idea is described in detail, but each idea is not evaluated independently. The experiment of sound separation is designed based on CLIPSep and the performance with this dataset is improved

**Strengths:**

1. the four-stage curation process is simple and easy to reproduce
2. the number of samples in the dataset is the biggest among those collected for audio-visual aligned dataset
3. the experiments using the dataset have shown improvements

**Additional Feedback:**

see above

**Documentation:**

No problem

**Ethics:**

no ethic concern

**Limitations:**

The limitation is described in a supplementary file

**Opportunities For Improvement:**

1. each stage of the curation process is not evaluated and it is difficult to know which one can be skipped. No ablation study for the threshold to classify non-corresponding vs corresponding
2. the performance boost of vision-queried sound separation is very limited. It is difficult to know if the improvement is thank to the dataset size or its correspondence quality

**Relation To Prior Work:**

sufficient

**Summary And Contributions:**

The paper proposes a new audio-visual dataset that offers a large amount of semantically aligned sounding videos. The dataset is created from Audio Set and Panda-70M and its value is proved by two applications: audio-visual retrieval and vision-queried sound separation

---

> ### Author Rebuttal · Authors · 2024-08-16
>
> Thank you for acknowledging the construction process and scale of our dataset, as well as affirming its value. We would like to address and clarify the specific questions you raised:
>
> **Q1: Thorough Validation of Critical Dataset Construction Steps**
>
> Our dataset construction process is structured into four key stages: (1) Data Collection, (2) Audio-Visual Correspondence Filtering, (3) Voice-Over Filtering, and (4) Sample Recycling with Sound Separation. Below is a brief overview of each stage:
>
> 1. **Data Collection:** This phase is focused on gathering data. To ensure broad applicability and relevance, we selected large-scale open-domain datasets like AudioSet [1] and Panda70M [2] as our primary data sources. These datasets already account for sample diversity, so additional diversity verification is not required at this stage. However, if alternative methods like web scraping are used, additional steps such as verifying sensitive information may be necessary.
> 2. **Audio-Visual Correspondence Filtering:** This is the most crucial stage, aimed at filtering out non-corresponding audio-video data samples. In Section 3.3, we employ another audio-visual representation learning model for cross-validation to demonstrate the effectiveness of our filtering methods. The results show that the audio-visual consistency of our filtered dataset surpasses that of VGGSOUND, underscoring the necessity and effectiveness of this step.
> 3. **Voice-Over Filtering:** This stage aims to filter out samples where the audio and video are only partially aligned, such as a video of a passing train whose audio includes both a train whistle and music.
> To address this, we use the PANNs [3] model, which has a sound classification AUC of 97.3%, to accurately identify whether a sample contains voice or music. Before applying this model, we randomly sampled 1,000 audio samples and manually verified the accuracy of the PANNs annotations. This step is crucial to ensuring that partially consistent samples are effectively filtered out.
> 4. **Sample Recycling with Sound Separation:** The aim of this stage is to further increase the scale of non-speech samples. We conducted extensive sampling and quality verification, with several samples showcased on our demo page to illustrate the high quality of sound separation. The results from this stage are then sent back to the second stage for additional quality verification. Note that this step can be omitted if dataset expansion is not required.
>
> **Q2: Impact of Audio-Visual Consistency and Dataset Scale on Model Performance**
>
> We address this issue through experiments E3, E4, and E5 in Table 6, which demonstrate the following:
>
> | # ID | Training Data | Data Scale | Mean SDR | Med SDR |
> | --- | --- | --- | --- | --- |
> | E3 | AudioSet | 1,445,360 | 5.004±0.103 | 6.781±0.164 |
> | E4 | AudioSet +AV-Correspondence Filtering | 898,366 | 5.646±0.101 | 7.682±0.162 |
> | E5 | AVSET-700K | 727,530 | 5.774±0.103 | 7.802±0.161 |
> | E8 | AVSET-200K | 200,000 | 5.152±0.103 | 6.928±0.168 |
>
> The experimental results show that although E3 contains more samples, the model performance is poor due to the lack of audio and video consistency in the data. After audio-visual consistency filtering, the Mean SDR of E4 improved by 0.642. After removing voice-over samples, the Mean SDR of E5 further improved by 0.128. Although the data scale of E4 and E5 is not as large as that of E3, better performance was achieved, highlighting the significance of audio-visual consistency for model performance.
>
> To further assess the impact of data scale on model performance, we randomly sampled 200K samples from AVSET-700K for experiments (E8). The performance dropped significantly, which demonstrates the importance of data scale. However, E8 still outperformed E3, proving that audio-visual consistency is more critical than data scale.
>
> Thank you again for your thoughtful feedback. We hope our response addresses your concerns, and we welcome further discussion at any time.
>
> [1] Gemmeke J F, Ellis D P W, Freedman D, et al. AudioSet: An ontology and human-labeled dataset for audio events[C]//2017 IEEE International Conference on Acoustics, Speech and Signal Processing (ICASSP). IEEE, 2017: 776-780.
>
> [2] Chen T S, Siarohin A, Menapace W, et al. Panda-70M: Captioning 70M videos with multiple cross-modality teachers[C]//Proceedings of the IEEE/CVF Conference on Computer Vision and Pattern Recognition. 2024: 13320-13331.
>
> [3] Kong Q, Cao Y, Iqbal T, et al. PANNs: Large-scale pretrained audio neural networks for audio pattern recognition[J]. IEEE/ACM Transactions on Audio, Speech, and Language Processing, 2020, 28: 2880-2894.

---

> > ### Author Rebuttal · Authors · 2024-08-19
> >
> > Dear Reviewer aXDx,
> >
> > Thank you for your positive reception of our work and your valuable comments. We appreciate the opportunity to address your questions and concerns.
> >
> > In our rebuttal, we outlined the necessity of each step in the dataset construction and detailed the verification processes for key components. Additionally, we conducted ablation experiments on data volume and sample correspondence quality, followed by a thorough analysis. We hope these efforts address your concerns. Your suggestions have significantly enriched our experimental analysis, and we are grateful for your input. If you have any further questions, please feel free to reach out at any time; we are eager to resolve any remaining doubts.
> >
> > Thank you once again for your valuable contribution.
> >
> > Sincerely,
> > Author of Paper 685

---

> > > ### Comment · Reviewer_aXDx · 2024-08-28
> > >
> > > Thanks for your detailed comments on one of my two concerns. How about your opinion on my second concern for the incremental performance boost of vision-queried sound separation?

---

> ### Author Rebuttal · Authors · 2024-08-28
>
> Thank you for acknowledging our response. We hope it further addresses your concerns:
>
> **Regarding the second concern:** *The performance boost in vision-queried sound separation appears limited. It remains challenging to discern whether this improvement is due to the dataset size or the quality of its correspondence.*
>
> **R2-1:** Regarding the limited performance improvement, we deliberately tested our method on VGGSOUND (some samples in it contain multiple sounds, which pose a significant challenge for all models compared in our paper to accurately separate) instead of the carefully selected VGGSOUND-CLEAN used in CLIPSEP, to validate the generalizability and the robustness of our vision-guided sound separation model. **The diverse and complex samples in VGGSOUND led to less pronounced improvements.** Here, we presented the performance comparison on VGGSOUND-CLEAN, the significant enhancements brought by our dataset become evident.
>
> | VGGSOUND-CLEAN (batchsize=128) | Mean SDR | MED SDR |
> | --- | --- | --- |
> | VGGSOUND | 5.76 | 5.65 |
> | AVSET-700K | **7.32** | **6.40** |
>
> **R2-2:** In our rebuttal (Q2), we analyzed this by comparing the original experimental results (E3, E4, E5) with additional experiments (E8) to determine if the model's effectiveness is driven by data scale or correspondence quality.
>
> | # ID | Training Data | Data Scale | Mean SDR | Med SDR |
> | --- | --- | --- | --- | --- |
> | E3 | AudioSet | 1,445,360 | 5.004±0.103 | 6.781±0.164 |
> | E4 | AudioSet +AV-Correspondence Filtering | 898,366 | 5.646±0.101 | 7.682±0.162 |
> | E5 | AVSET-700K | 727,530 | **5.774±0.103** | **7.802±0.161** |
> | E8 | AVSET-200K | 200,000 | 5.152±0.103 | 6.928±0.168 |
>
> Our findings are as follows:
>
> - **E3 vs. E4:** Consistency filtering significantly enhances sound separation, with a Mean SDR improvement of 0.642.
> - **E4 vs. E5:** Filtering out irrelevant voice-overs further boosts performance, with a Mean SDR increase of 0.128.
> - **E5 vs. E8:** Dataset size notably impacts performance. Training with fewer samples results in a marked performance drop.
> - **E3 vs. E8:** Even with limited samples, the model trained on AVSET-200K, which has higher audio-visual consistency, outperforms the model trained on AudioSet (1.5M).
>
> In summary, we believe that audio-visual consistency is the most critical factor for this task. While maintaining high audio-visual consistency, expanding the dataset size is also essential for enhancing model performance—both are indispensable.
>
> Thank you again for your feedback and suggestions, which have significantly improved the adequacy of our experiment. If you have any further questions, please feel free to reach out to us anytime.

---

> > ### Comment · Reviewer_aXDx · 2024-08-28
> >
> > Thanks for addressing all my concerns. I think it's good to include your analysis in the above comments in the revised paper. Anyway, I raised my score

---

> > > ### Author Response · Authors · 2024-08-29
> > >
> > > Thank you for recognizing our work and improving the score! We also appreciate your valuable suggestions, which we will incorporate into the final version. If you have any further questions, we welcome continued discussion at any time.

---

### Author Rebuttal · Authors · 2024-08-16

We sincerely thank the reviewers for their insightful feedback, which has greatly contributed to the improvement of our work. We value the time and effort put into reviewing our submission and are committed to incorporating all feedback into the final version of our paper. Below is a summary of our key contributions and ethical considerations:

# [Contributions]

We are pleased to highlight the major contributions of our AVSET-10M dataset to the research community:

- **Comprehensive Large-Scale Audio-Visual Dataset**: AVSET-10M extends large-scale open-source datasets like AudioSet and Panda70M, providing an extensive repository of audio-visual data that supports diverse research and development efforts across multiple domains.
- **Extensive and Diverse Audio Categories**: The dataset covers a wide range of audio categories, offering a valuable resource for training and evaluating models in tasks such as sound recognition, separation, and classification, catering to various research needs.
- **High Audio and Video Consistency**: AVSET-10M emphasizes strong audio and video synchronization, making it an ideal dataset for studies requiring consistent and reliable audio-visual data.
- **Robust Dataset Construction and Verification**:
    - We have meticulously designed a threshold selection scheme to ensure the quality and relevance of the data, and rigorously verified its effectiveness.
    - The voice-over filtering stage filters out samples where the audio and video are only partially consistent, further enhancing the audio-visual consistency.
    - Our innovative sample recycling method, utilizing sound separation, significantly increases the availability of samples beyond just speech or music.

These contributions demonstrate our commitment to advancing audio-visual research. We believe AVSET-10M will be a valuable resource for the community, and we welcome further feedback.

# [Ethical Considerations]

We appreciate the reviewers’ concerns regarding the ethical aspects of our dataset and are grateful for the opportunity to address them:

- **Privacy Concerns**: AVSET-10M is constructed from existing open-source datasets and contains only links to videos, not the actual content. To address privacy concerns, we have implemented a removal request mechanism, allowing individuals to request the deletion of links to privacy-sensitive content. Acknowledging the limitations of user-initiated requests, we plan to regularly update our sample repository from upstream datasets like AudioSet and Panda70M, proactively identifying and removing any videos that raise privacy concerns, ensuring ongoing compliance with privacy standards.
- **De-identification**: While our dataset does not include personal identifiers, we recognize that audio data could still pose security risks. To mitigate these risks, we are committed to exploring and implementing de-identification techniques, such as assessing speech recognition results to remove data containing personal information.
- **Population Representativeness**: We appreciate the reviewer’s insight into the potential population bias arising from our reliance on upstream datasets without further population analysis. While YouTube videos are sourced globally, offering some degree of representativeness, privacy protections make it challenging to determine the exact locations of the videos, complicating in-depth population analysis. We will acknowledge this limitation in future revisions of our paper.
- **Broader Ethical Considerations**: Beyond privacy, we are dedicated to addressing other ethical concerns, such as deception and harassment. We will explore and implement de-identification techniques when handling personal voice data to minimize the risks of identity exposure and harassment. Furthermore, we will establish clear terms of use and ethical guidelines to prevent the misuse of our dataset for deceptive or harassing purposes.

We are committed to revising our paper to reflect all of these considerations and look forward to the opportunity to submit a stronger final version. Thank you once again for your invaluable feedback.

---

### Author Rebuttal · Authors · 2024-08-21

Dear AC and Reviewers,

Thank you again for the great efforts and valuable comments. We have carefully addressed the main concerns in detail. We hope you might find the response satisfactory. As we are currently in the discussion stage, **we are very much looking forward to hearing from you about any further feedback.** We will be very happy to clarify any further concerns (if any).

Best, Authors

---

### Author Response · Authors · 2024-09-01
**Summary of Contributions and Revisions.**

## [Our Contributions]

We are glad to find out that the reviewers generally acknowledge our contributions:

1. **Contribution**
    - The significant importance of the audio-visual consistency dataset. **[zYSb, hCbp, T2Qw]**
    - The large scale of the dataset. **[aXDx, zYSb, hCbp]**
2. **Soundness**
    - The well-designed dataset process and assured data quality. **[aXDx, zYSb, hCbp]**
    - The well-planned experimental design. **[zYSb, hCbp]**
3. **Presentation**
    - The dataset construction process is clearly described and easily understandable. **[aXDx, hCbp]**
    - The paper is presented in a clear manner. **[aXDx, hCbp, T2Qw]**
    - Substantial efforts have been made to protect privacy and address ethical issues. **[Ethics Reviewer acXv, Ethics Reviewer SCjW, aXDx, zYSb, hCbp]**

## [New Experiments and Revisions]

As the discussion phase comes to an end, we would like to review the revisions we have made to the manuscript based on the reviewers' suggestions, which have significantly improved the quality of our work:

1. **New Experiments:**
    - Analysis of the impact of audio-visual consistency and dataset scale on performance. **[aXDx]**
    - Comparison and analysis of model performance on VGGSOUND-CLEAN. **[aXDx]**
    - Experiments comparing and analyzing the inference speed of using sound separation methods for voice-over detection. **[zYSb]**
    - Comparison and analysis of similarity with and without voice-over. **[T2Qw]**
2. **Textual Descriptions:**
    - Changed "*abstract audio scenes*" to "*audio events that cannot be identified by unique visual content*" and added examples for better understanding. **[zYSb]**
    - Improved the expression regarding the 68–95–99.7 rule to be more precise. **[zYSb]**
    - Maintained a neutral and objective tone and supplemented publication information. **[zYSb, T2Qw]**
    - Added details about the construction of AudioSet. **[T2Qw]**
    - Further refined the experimental details on vision-guided sound separation. **[T2Qw]**
    - Clarified the verification of non-overlapping train/test sets. **[T2Qw]**
    - Further elaborated on the dataset construction process with examples for clarity. **[T2Qw,** **zYSb, aXDx]**
    - Discussed the application of the dataset to additional tasks. **[hCbp]**
    - Added a discussion on the potential lack of diversity due to reliance on a single data source in the limitations section. **[hCbp]**
    - Added a discussion on the dependency of the filtering process on multimodal embeddings and the cross-validation method used to address this issue in the limitations section. **[zYSb]**
3. **Ethical Considerations:**
    - Added a discussion limitations related to dataset privacy issues and new measures (regular proactive updates to remove lists, de-identification). **[hCbp, T2Qw, Ethic Reviewer acXv, Ethic Reviewer SCjW]**
    - Added a discussion on population representativeness in the limitations section. **[Ethic Reviewer acXv]**

Finally, we would like to thank the AC and all the reviewers once again. Your suggestions have greatly enhanced the quality of this paper! We wish you all success in your research endeavors!

---

### Decision · Program_Chairs · 2024-09-26

**Decision:**

Reject

**Comment:**

The paper contributes an dataset, which is valuable for the community. However, it raises several ethical concerns. One proposed mitigation strategy is to provide links to videos rather than including the videos directly. This approach is problematic as a benchmark, since these links may become invalid over time, compromising the dataset's reliability as a reference for future research. Given these issues and the impact on the benchmark's robustness, I am inclined to recommend rejecting the paper at this stage.